# IDEST: Assessing Self-Supervised Learning Representations via Intrinsic Dimension

**Julie Mordacq** [1 2]   **Vicky Kalogeiton** [2]   **Steve Oudot** [1 2]

## Abstract

Self-supervised learning (SSL) has emerged as a powerful paradigm for learning meaningful representations from unlabeled data. However, the standard protocol for evaluating these representations, linear probing, is computationally expensive, sensitive to hyperparameters, and provides limited insight into the geometric structure of the representation space. In this work, motivated by connections between neural network generalization and intrinsic dimension (ID) we propose **IDEST**, a method for estimating the ID of SSL representations via the Minimum Spanning Tree dimension estimator ($\dim_{\mathrm{MST}}$). Across diverse datasets, architectures, and SSL pretraining objectives, we show that **IDEST** strongly correlates with downstream linear probe performances. Furthermore, we demonstrate that **IDEST** enables efficient hyperparameter selection, significantly reducing the computational cost compared to supervised alternatives. Our results highlight intrinsic dimensionality as a principled geometric proxy for assessing SSL representations, complementing standard supervised probing protocols.

## 1. Introduction

*Can the geometry of learned representations provide reliable, label-free insights into their quality for downstream tasks?* Self-supervised learning (SSL) offers a natural setting to ask this question, as its objectives are explicitly designed to structure representations without access to labels (Chen et al., 2020; Bardes et al., 2022; Assran et al., 2023; Venkataramanan et al., 2025; Siméoni et al., 2025; Mordacq et al., 2025). Beyond their strong performance, SSL representations are valued for their ability to transfer to a wide range of downstream tasks with minimal super-

vision (Dufour et al., 2025; Couairon et al., 2025; Maruani et al., 2025; Degeorge et al., 2025). Rather than optimize task-specific decision boundaries, SSL methods shape the organization of data in the representation space through geometric constraints such as alignment between views, feature uniformity, and variance control. As a result, intrinsic properties of the resulting manifold, e.g., curvature and spectral structure, have emerged as potential indicators of representation quality, prompting recent efforts to investigate such geometric proxies (Ansuini et al., 2019; Garrido et al., 2023).

Among them, intrinsic dimension (ID), originally introduced by Bennett (1969), has emerged as a particularly informative quantity: it characterizes the effective number of degrees of freedom required to represent data in an embedding space. Recent studies have revealed a monotonic relationship between a model's generalization error and the intrinsic dimension of its representations (Ansuini et al., 2019; Konz & Mazurowski, 2024), with lower intrinsic dimension often correlating with improved downstream accuracy. These findings, grounded in the *manifold hypothesis* (Goodfellow et al., 2016), suggest that representation quality is governed not only by separability, but also by how efficiently information is compressed into low-dimensional geometric structure. Yet, existing evidence is largely confined to supervised convolutional networks, leaving open how intrinsic dimension behaves across modern self-supervised methods and whether it indicates downstream performance in this setting.

In practice, estimating ID is far from trivial. Intrinsic dimension admits multiple mathematical formalizations, topological, fractal, and information-theoretic, each capturing different facets of data geometry, and since ID cannot be observed directly but must be estimated from finite samples, the choice of estimator matters. Most notably, nearest-neighbor-based methods such as TwoNN (Facco et al., 2017) and maximum likelihood estimators (Levina & Bickel, 2004) have been widely adopted, but suffer from well-known limitations: They rely on strong locality and isotropy assumptions, are sensitive to noise and finite-sample effects, and become unstable in high-dimensional or highly structured representation spaces. These limitations are especially pronounced in SSL, which requires operating far from standard

---

[1]Inria Saclay [2]LIX, CNRS, École Polytechnique, IP Paris. Correspondence to: Julie Mordacq <julie.mordacq@inria.fr>.

*Proceedings of the $43^{rd}$ International Conference on Machine Learning*, Seoul, South Korea. PMLR 306, 2026. Copyright 2026 by the author(s).

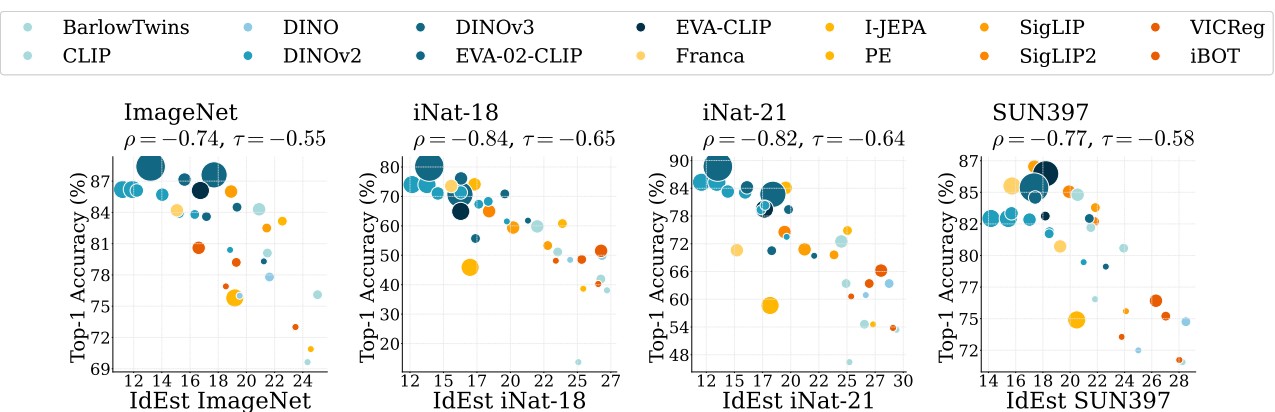

*Figure 1.* **Foundation Models and IDEST: Intra-Dataset Correlation.** Linear probe accuracy of pretrained SSL models on ImageNet (left), iNat-18 (middle left), iNat-21 (middle right), SUN397 (right) versus IDEST on each respective dataset. Each point corresponds to a model checkpoint; point size reflects the number of parameters. We report Kendall's $\tau \in [-1, 1]$ and Spearman's $\rho \in [-1, 1]$. Correlations across all four benchmarks demonstrate IDEST's ability to provide insights into models' representation quality.

conditions. First, it operates in a non-asymptotic regime where $n \approx d$ rather than $n \to \infty$ with $d$ fixed, where $n$ is the number of samples and $d$ is the ambient dimension. Second, SSL objectives introduce dependencies between data points, violating standard independence assumptions. For instance, as shown in Figure 2, TwoNN and MLE are sensitive to the latter; they are unable to reliably detect the intrinsic dimension, with TwoNN even diverging.

In this paper, we consider a complementary ID estimator grounded in the scaling behavior of minimum spanning trees (Costa & Hero, 2006). Asymptotically tied to the intrinsic Rényi entropy, this estimator balances local and global information, enjoying robustness to noise, to sampling-density variations, and to high ambient dimensionality. Furthermore, its reliability under sampling sparsity, a behavior consistent with known theoretical results on MST length in sparse regimes (e.g., Theorem 1 in (Mordacq et al., 2025)), makes it particularly well-suited to the $n \approx d$ regime inherent in vision SSL.

Building on this perspective, we propose IDEST (pronounced as *id est*, Latin for 'that is'), standing for ID Estimation for SSL using minimum spanning Tree. IDEST is an unsupervised criterion for evaluating self-supervised representations based on intrinsic dimension estimation via minimum spanning trees. We show that IDEST strongly correlates with downstream performance across a wide range of self-supervised objectives, including joint-embedding, joint-predictive and vision-language alignment (Figures 1, 3 and 4). Moreover, IDEST provides an efficient proxy to supervised linear probing, enabling practical hyperparameter selection without labels at a fraction of the computational cost of supervised probing.

Our main contributions can be summarized as follows:

*i)* We propose IDEST, an unsupervised criterion based on the intrinsic dimension of the representation (Section 3).

*ii)* We demonstrate that IDEST serves as an efficient proxy to assess the quality of SSL representations (Section 4.1).

*iii)* We show that IDEST enables unsupervised hyperparameter selection across diverse SSL objectives and architectures (Sections 4.2 and 4.3).

Beyond providing empirical insights, our work underscores the promise of geometric descriptors as a complement to traditional supervised evaluation methods.

## 2. Related Work

Our work builds on recent efforts to analyze geometric properties of deep neural networks (DNNs) in a label-free setting, and to understand their connection to generalization. Existing literature investigates these properties through two main lenses: spectral properties and intrinsic dimensionality.

### 2.1. Representation Spectrum

$\alpha$-ReQ (Agrawal et al., 2022) and RankMe (Garrido et al., 2023) characterize the eigenspectrum of representations within SSL frameworks, by measuring the decay rate of empirical eigenvalues or calculating the effective rank of the representations, respectively. These studies showed that both metrics often strongly correlate with downstream performance and highlighted their utility for hyperparameter selection.

However, the power-law assumptions underlying $\alpha$-ReQ are violated in the presence of representation collapse (He & Ozay, 2022) (where networks output identical or non-

informative vectors regardless of the input, resulting in rank-deficient representations), leading to unreliable performance estimates when the embedding space becomes rank-deficient (Garrido et al., 2023). Furthermore, RankMe is limited to the study of Joint-Embedding Architectures (JEAs) where two networks are trained to produce similar embeddings for different views of the same image. Notably, RankMe is tailored to identify a pivotal challenge of JEAs: representation collapse (Jing et al., 2021). This narrow focus potentially limits the applicability of such metrics to other SSL paradigms less prone to this specific type of collapse. For instance, as shown in Table 1, RankMe is less effective on I-JEPA, a joint-predictive method.

More recently, Thilak et al. (2024) proposed LiDAR, which quantifies the rank of the Linear Discriminant Analysis matrix associated with the surrogate SSL task, a measure that intuitively captures how much information the representation retains for solving that task. Crucially, LiDAR leverages SSL pretraining information (i.e., augmentations), whereas our work targets the harder setting, where only frozen representations are accessible, without any access to training data or pairing structure, aiming to gain a deeper understanding of SSL models beyond hyperparameter selection.

### 2.2. Intrinsic dimension

Intrinsic dimension (ID) has been linked to the generalization of DNNs through two primary lenses: *the optimization trajectory*, analyzing the sequence of model states and parameter updates during training (Simsekli et al., 2020; Birdal et al., 2021; Dupuis et al., 2023; Tan et al., 2024), and *the learned representations* (Ansuini et al., 2019; Konz & Mazurowski, 2024; Ruppik et al., 2025). In this work, we focus on the latter. Analyzing optimization trajectories is often computationally expensive, as it requires processing all network parameters (which can reach hundred of millions in current vision models, e.g., ViT-L (Dosovitskiy et al., 2021)) and necessitates access to intermediate training checkpoints that are rarely available for large-scale, pre-trained models.

Regarding representational analysis, several studies have investigated the ID of Large Language Model (LLM) representations (Aghajanyan et al., 2021; Cai et al., 2021; Tulchinskii et al., 2023; Valeriani et al., 2023; Viswanathan et al., 2025; Lee et al., 2025; Ruppik et al., 2025) demonstrating that ID provides critical insights into training dynamics and generalization. However, research on computer vision models has largely remained restricted to supervised convolutional neural networks (CNNs). For instance, Ansuini et al. (2019) showed that the ID of supervised CNNs correlates with performance, while Konz & Mazurowski (2024) proposed a generalization scaling law based on representational ID, though their validation was limited to supervised CNN architectures.

Despite these advances, the use of intrinsic dimension estimations to characterize representation quality remains largely unexplored in self-supervised learning.

## 3. Generalization and Dimension Estimation

This section first introduces the theoretical connection between intrinsic dimension and generalization (Section 3.1). Second, it presents standard intrinsic dimension estimators used in prior studies of deep neural network representations, along with their limitations, and motivates the use of an alternative: $\dim_{\text{MST}}$ (Section 3.2).

### 3.1. Theoretical connection to generalization

Consider a classification dataset $\mathcal{D}$, consisting of $N_D$ points $x \in \mathbb{R}^n$ with target labels $y = \mathcal{F}(x)$ defined by an unknown function $\mathcal{F} : \mathbb{R}^n \to \mathbb{R}^C$, where $C$ is the number of classes. The dataset is split into a training set $\mathcal{D}_{\text{train}}$ and a test set $\mathcal{D}_{\text{test}}$. We analyze 'well-trained' models $f : \mathbb{R}^n \to \mathbb{R}^C$ that interpolate the training data, such that $f(x) = \mathcal{F}(x)$ for all $x \in \mathcal{D}_{\text{train}}$. Let $\mathcal{L}$ be a non-negative loss function (e.g., cross-entropy) satisfying $\mathcal{L}(f(x), \mathcal{F}(x)) = 0$ if $f(x) = \mathcal{F}(x)$. The generalization error is expressed as the expected loss over $\mathcal{D}_{\text{test}}$. The model can be decomposed as $f = h \circ g$, where $g$ is an encoder (e.g., a pre-trained SSL backbone) that produces latent representations living on some $d$-dimensional manifold, and where $h$ is a classification head.

**Theorem 3.1.** *Konz & Mazurowski (2024) Let $K_L$ the Lipschitz constant of the loss function. Then:*

$$\mathcal{L} \simeq \mathcal{O}\left(K_L N_D^{-1/d}\right) \tag{1}$$

This relation suggests that, for a fixed dataset size $N_D$, the error is dominated by the $-\frac{1}{d}$ exponent. Consequently, a lower $d$ implies a more efficient representation. We therefore use the estimated ID of the representations as an unsupervised criterion to assess the quality of the downstream task.

### 3.2. Intrinsic dimension estimators

The intrinsic dimension (ID) estimation problem assumes the data points are sampled on (or close to) some unknown $d$-submanifold of the ambient space. The goal is to estimate $d$ from the data.

#### 3.2.1. PARAMETRIC ESTIMATORS

While many estimators exist (Johnsson et al., 2014; Tempczyk et al., 2022; Binnie et al., 2025), two main estimators have been adopted in prior studies of deep neural network representations (Ansuini et al., 2019; Pope et al., 2021; Konz & Mazurowski, 2024; Ruppik et al., 2025):

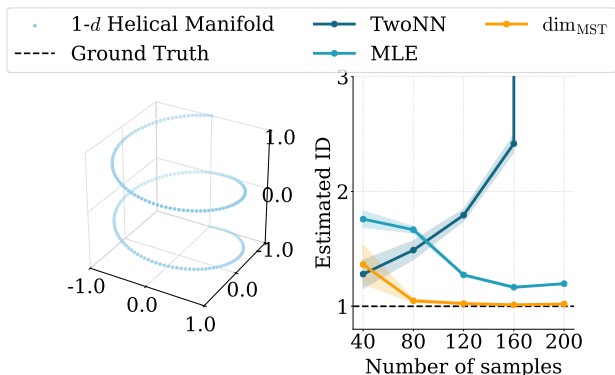

*Figure 2.* **Impact of Sampling Distribution on Estimators.** *(Left)* 200 points sampled evenly along a 1-dimensional helix. *(Right)* Estimated ID as a function of sample size. While $\dim_{\text{MST}}$ converges accurately to the ground truth $d = 1$ as the sample size increases, MLE and TwoNN do not, with TwoNN even diverging to infinity.

*Maximum Likelihood Estimation (MLE)* (Levina & Bickel, 2004) and *TwoNN* (Facco et al., 2017). Both are parametric estimators, founded on the assumption that the data points are sampled i.i.d. from a probability distribution supported on the submanifold, with locally constant density, and both treat the data points locally as a homogeneous Poisson process. Under these conditions, the number of points within a small $\varepsilon$-ball, and the ratio between the distances to the second and first nearest neighbors, follow specific parametric distributions whose parameters depend directly on $d$ and can be inferred, either via maximum likelihood estimation for MLE or via regression for TwoNN.

These methods are inherently tied to specific data distributions, and they become unstable as the input data deviate from those, even in simple cases. Consider, for instance, the sample of Figure 2, composed of up to 200 points evenly spaced along a 1-dimensional helix embedded in $\mathbb{R}^3$. Despite the high regularity of both the submanifold and the sample, both methods fail to recover the correct intrinsic dimension, with TwoNN even diverging to infinity in the asymptotic regime. This behavior is consistent across standard choices of hyperparameters of the methods. In practice, this instability translates into a degradation of the performances in the presence of noise (Tulchinskii et al., 2023; Binnie et al., 2025), or on heavy-tailed distributions (Birdal et al., 2021).

### 3.2.2. METRIC INVARIANTS BASED ESTIMATORS

The limitations of the parametric methods TwoNN and MLE motivate a shift toward estimators based on the theory of Euclidean functionals (Yukich, 2006). In particular, the asymptotic growth rate of the length of Minimum Spanning Tree (MST) are related to the Rényi entropy of the

underlying distribution. This connection has enabled the derivation of several dimension estimators with proven consistency under relatively weak assumptions: compactness of the manifold and boundedness of the Lebesgue sampling density supported on the manifold (Costa & Hero, 2006). For instance, the *Minimum Spanning Tree dimension estimator*, or $\dim_{\text{MST}}$ for short, successfully recovers the ground truth dimension $d = 1$ in the example of Figure 2.

Given a point cloud $Z$ in $\mathbb{R}^D$, the Minimum Spanning Tree (MST) is the acyclic connected graph $G = (V, E)$, with vertex set $V = Z$, that minimizes the total edge length:

$$L(G) = \sum_{(z,z') \in E} \|z - z'\|_2. \tag{2}$$

Costa & Hero (2006) studied the growth rate of the length of the minimum spanning tree for random point clouds in Riemannian manifolds. Given an i.i.d. $n$-sample $X_n$ drawn from a fixed probability measure $P_X$ supported on a compact Riemannian $d$-manifold $M$ with density $f_X$ w.r.t. the Hausdorff measure, there exists a constant $C'$ independent of $f_X$ and of $M$ such that, almost surely:

$$n^{-\frac{d-1}{d}} \cdot L\left(\text{MST}(X_n)\right) \xrightarrow[n \to \infty]{} C' \int f_X^{\frac{d-1}{d}} \, \mathrm{d}\mathcal{H}, \tag{3}$$

where $\mathcal{H}$ denotes the Hausdorff measure on $M$.

This result motivates the definition of $\dim_{\text{MST}}$ given by Costa & Hero (2006):

**Definition 3.2.** Given a bounded metric space $M$, the MST dimension of $M$, denoted by $\dim_{\text{MST}}(M)$, is the infimal exponent $d \in \mathbb{N}$ such that $L\left(\text{MST}(X)\right)/|X|^{\frac{d-1}{d}}$ is uniformly bounded for all finite subsets $X \subseteq M$:

$$\dim_{\text{MST}}(M) := \inf\Big\{d : \exists C \text{ s.t. } \frac{L\left(\text{MST}(X)\right)}{|X|^{\frac{d-1}{d}}} \leq C$$
$$\text{for every finite subset } X \text{ of } M\Big\}.$$

In practice, the ID is estimated via log-log linear regression (Birdal et al., 2021; Binnie et al., 2025). Specifically, given subsamples $X_{n_i}$ with increasing sizes $n_i$, we fit:

$$\log(L(\text{MST}(X_{n_i}))) \approx \frac{d-1}{d} \log(n_i) + \log(C).$$

The resulting slope $m$ yields the intrinsic dimension via the relation $d = 1/(1 - m)$. The complete algorithm for computing $\dim_{\text{MST}}$ is given in Algorithm 1.

**Persistent Homology Dimension.** In Topological Data Analysis (TDA), the MST relates to the so-called *total degree-0 persistence of the Rips filtration* (Oudot, 2015). This connection allows for the definition of the *0-dimensional Persistent Homology (PH) dimension*, $\dim_{\text{PH}}^0$,

which is equivalent to the $\dim_{MST}$ (Adams et al., 2020). The PH dimension has been used in several dimension-estimation applications (Birdal et al., 2021; Dupuis et al., 2023). This connection provides further theoretical grounding for the robustness of $\dim_{MST}$. In particular, TDA-based measures such as the total persistence of the Rips filtration are provably stable under pertubations of the underlying distribution (Chazal et al., 2014). This was further observed empirically by Tulchinskii et al. (2023) in the context of LLM latent spaces.

## 4. IDEST: $\dim_{MST}$ for Unsupervised Assessment of Self-Supervised methods

In this section, we apply $\dim_{MST}$ to estimate the intrinsic dimension of self-supervised representations. This yields **IDEST**, which stands for Intrinsic Dimension Estimation for SSL using minimum spanning Trees. Our goal is to determine whether **IDEST** can yield unsupervised insights into downstream performances. Specifically, to empirically validate **IDEST**, we compare it against linear probing, the standard evaluation protocol for self-supervised learning (SSL).

Subsequently, we address the following research questions:

*Q1.* To what extent does **IDEST** reflect representation quality across pretrained SSL models?

*Q2.* Can **IDEST** yield insight along self-supervised pretraining?

*Q3.* Can **IDEST** be leveraged as a principled proxy for hyperparameter selection without requiring labeled data?

**Implementation of IDEST**

To satisfy the assumptions of Theorem 3.1, **IDEST** operates on the frozen representation passed to the classifier head, following each method's standard evaluation protocol (i.e., consistent with how linear probes are trained). For models without a class token (e.g., I-JEPA), we average-pool the patch tokens; for models with a `[CLS]` token (e.g., DINO, DINOv2), we use this token directly. Additional implementation details are provided in Appendix A.

### 4.1. IDEST and accuracy of foundation models

We begin by evaluating whether our proposed **IDEST** reflects representation quality across a diverse set of pretrained SSL models. Specifically, we compute **IDEST** on frozen representations and compare it against standard linear probing accuracy. We compute two rank correlation statistics: *Spearman's rank correlation coefficient $\rho$* (Spearman, 1961) and *Kendall's rank correlation coefficient $\tau$* (Kendall, 1938). Both capture monotonic relationships between rankings.

**Setup.** We evaluate 14 SSL methods spanning four

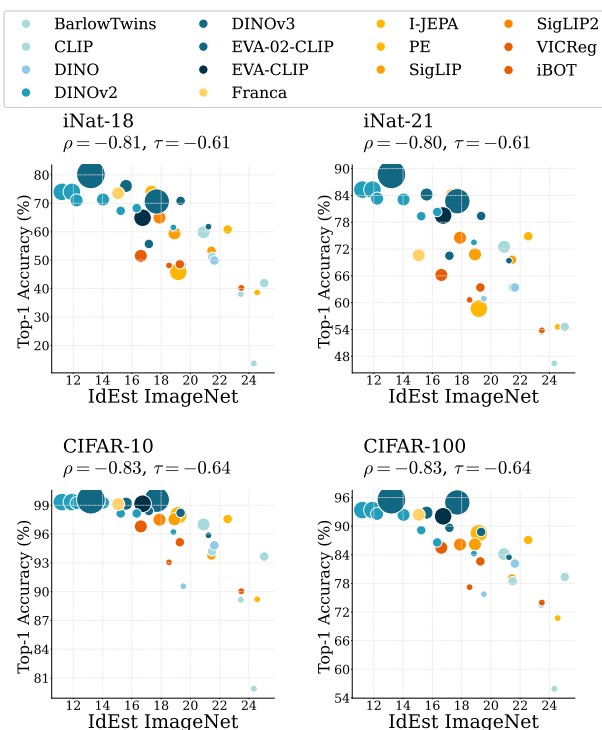

*Figure 3.* **Foundation Models and IDEST: Inter-Dataset Correlation.** Linear probe accuracy of pretrained SSL models on iNat-18 (top right), iNat-21 (top left), CIFAR-10 (bottom right), CIFAR-100 (bottom left) versus **IDEST** computed on ImageNet. Strong correlations demonstrate that **IDEST** computed on a single reference dataset: ImageNet, is indicative of model quality across datasets.

paradigms: pure joint-embedding (e.g., VICReg (Bardes et al., 2022), DINO (Caron et al., 2021)), joint-predictive (e.g., I-JEPA (Assran et al., 2023)), combined objectives (e.g., iBOT (Zhou et al., 2022), DINOv2 (Oquab et al., 2023)), and vision-language alignment (e.g., CLIP (Radford et al., 2021), EVA-CLIP (Sun et al., 2023)).

For each method, we evaluate two main architectures: ResNet (He et al., 2016) and ViT (Dosovitskiy et al., 2021), and we include various model scales (e.g., ViT-S, and ViT-G). This results in 33 different models. Table 3 (in supplementary) provides the complete list of models and architectures studied.

**Results.** We first examine Intra-Dataset Correlation in Figure 1, where **IDEST** is estimated on each dataset separately and compared against the corresponding linear probe accuracy. Figure 1 reports results on ImageNet (Deng et al., 2009) and three additional fine-grained datasets: iNat-18 (Van Horn et al., 2018), iNat-21 (Van Horn et al., 2021), and SUN397 (Xiao et al., 2010). Each point represents a pretrained SSL model, spanning diverse architectures, training objectives, and scales.

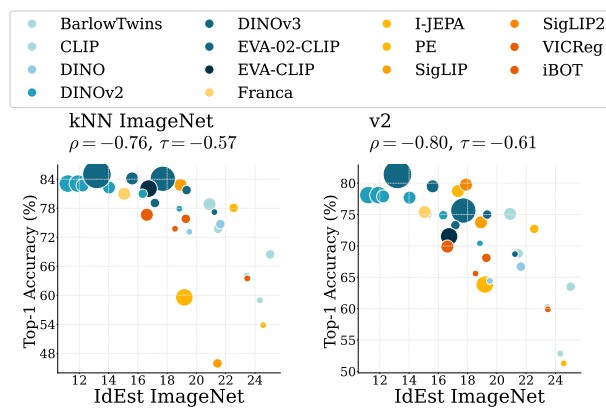

| | | | |
|---|---|---|---|
| ● BarlowTwins | ● DINOv3 | ● I-JEPA | ● SigLIP2 |
| ● CLIP | ● EVA-02-CLIP | ● PE | ● VICReg |
| ● DINO | ● EVA-CLIP | ● SigLIP | ● iBOT |
| ● DINOv2 | ● Franca | | |

*Figure 4.* **Foundation Models and IDEST: Alternative Evaluation Protocol**. Accuracy under alternative evaluation settings versus **IDEST**. Strong correlations demonstrate that **IDEST** computed on a single reference dataset: ImageNet, is indicative of model quality across evaluation protocols.

Across all datasets, Figure 1 reveals a consistent negative correlation between **IDEST** and downstream linear probing accuracy: models with lower intrinsic dimension tend to achieve higher performance. This trend holds on ImageNet (left) as well as on the fine-grained benchmarks iNat-18 (middle left), iNat-21 (middle right), and SUN397 (right). Notably, the relationship is consistent across a wide range of SSL paradigms, joint-embedding methods, joint-predictive methods, and vision-language pretraining, suggesting that intrinsic dimension acts as a unifying geometric descriptor of representation quality, independent of the training objective.

This is further supported by the correlation metrics reported atop each plot. Both Kendall's $\tau \approx -0.6$ and Spearman's $\rho \approx -0.8$ confirm that **IDEST** reliably preserves the relative ordering of models across all four datasets, in strong agreement with linear probing rankings.

While our analysis has first focused on Intra-Dataset Correlation, a natural question arises: do these findings persist when **IDEST** and accuracy are measured on *different* datasets, or under alternative evaluation protocols?

In Figure 3, we study Inter-Dataset Correlation, where **IDEST** is computed on ImageNet and accuracy is evaluated on four target datasets: the large-scale fine-grained benchmarks iNat-18 and iNat-21, and the smaller-scale datasets CIFAR-10 and CIFAR-100. The correlation remains strong across both target datasets, suggesting that **IDEST** captures intrinsic properties of the learned representations rather than dataset-specific characteristics.

In Figure 4, we examine **IDEST** under Alternative Evaluation Protocols. **IDEST** remains in-

dicative of performance on ImageNet under kNN evaluation, and on the complementary ImageNet-v2 validation set.

Overall, these results reveal that low intrinsic dimension is a consistent feature of well-performing representations, and suggest that **IDEST** serves as a simple, label-free proxy for downstream evaluation (Figures 1, 3 and 4), offering insight into the quality of self-supervised representations without annotated data.

> **Finding 1. IDEST negatively correlates with linear probing accuracy:** low intrinsic dimension is a consistent geometric signature of strong representations across intra- and inter-dataset settings and alternative evaluation protocols.

### 4.2. Training dynamics: offline and online probing

Here, we study whether **IDEST** tracks representation quality during unsupervised training. To this end, we consider two complementary evaluation protocols: *(i) offline linear probing* and *(ii) online probing*. While both aim to assess downstream performance over the course of self-supervised pretraining, they differ in when and how the classifier is trained.

**Offline linear probing.** Offline linear probing is the standard evaluation protocol in self-supervised learning. The representation model is first trained without labels, after which a linear classifier is trained on frozen features. To analyze training dynamics, we extract multiple checkpoints during pretraining and perform linear probing independently for each checkpoint.

Figure 5 (top) reports results for VICReg (Bardes et al., 2022) (5a), DINO (Caron et al., 2021) (5b) and I-JEPA (Assran et al., 2023) (5c), respectively on ImageNet. We show linear probing top-1 accuracy (y axis on the left, in orange) as a function of training epochs, together with **IDEST** (y axis on the right, in blue). Across both models, **IDEST** closely follows the evolution of downstream accuracy throughout training. As representations improve and linear probing accuracy increases, the intrinsic dimension consistently decreases. This strong temporal correlation indicates that **IDEST** captures meaningful geometric changes in the representation space as training progresses, without requiring labels.

**Online linear probing.** We consider online probing, where a linear classification head is attached to the representation and trained jointly during self-supervised pretraining. Importantly, gradients from the classifier do not backpropagate into the representation encoder, ensuring that the learned features remain purely self-supervised. This setting allows

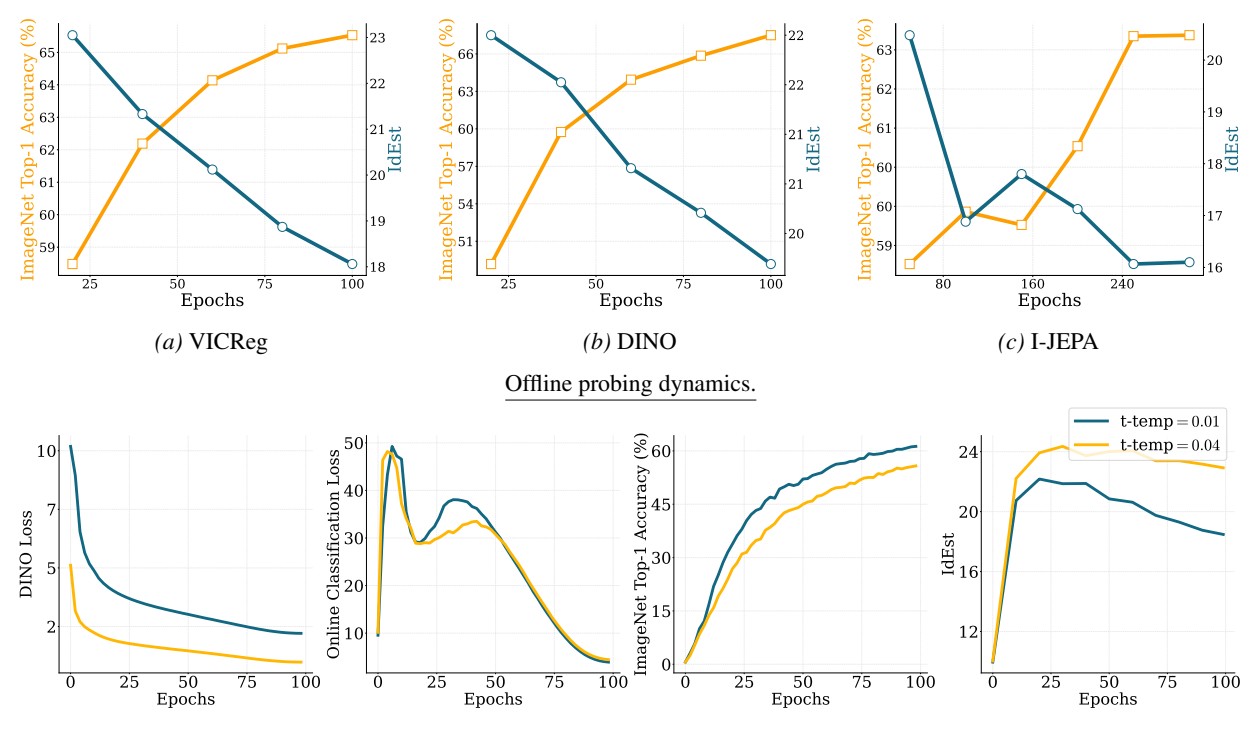

*(a)* VICReg         *(b)* DINO         *(c)* I-JEPA

Offline probing dynamics.

Online probing dynamics during DINO pretraining.

*Figure 5.* **Tracking Training Dynamics.** *(Top) Offline probing dynamics:* Evolution of ImageNet-1k linear probing top-1 accuracy (left y-axis, in orange) and **IDEST** (right y-axis, in blue) over self-supervised pretraining epochs for (a) VICReg (ResNet-50), (b) DINO (ViT-S), and (c) I-JEPA (ViT-B). As representations improve and linear probing accuracy increases, **IDEST** consistently decreases, demonstrating its ability to capture meaningful geometric changes during the evolution of the representation. *(Bottom) Online probing dynamics during self-supervised training:* Evolution of the self-supervised loss, classification loss, ImageNet-1k online classification top-1 accuracy, and **IDEST** over training epochs for DINO. While early-stage representations are highly constrained, **IDEST** progressively tracks improvements in downstream accuracy as training proceeds.

us to monitor downstream performance continuously during training.

Figure 5 (bottom) illustrates the training dynamics for DINO using a ViT-S, reporting the self-supervised loss (left), online classification loss and ImageNet top-1 accuracy (middle), and **IDEST** (right). We observe that during the early stages of training, particularly within the first 10 epochs, representations are highly constrained, rendering **IDEST** less informative. However, as training progresses, the self-supervised loss decreases and classification accuracy improves; concurrently, **IDEST** systematically drops, closely tracking the evolution of the other metrics. This demonstrates that **IDEST** faithfully reflects representation quality even in an online setting, capturing improvements in downstream performance as they emerge during the pretraining.

Overall, we observe that across both offline and online protocols, and across diverse SSL objectives, **IDEST** mirrors the evolution of downstream classification performance. These results further support its utility as a label-free indicator of representation quality, effective not only at convergence but throughout the pretraining process, once the initial training

stages are surpassed. This leads us to our second finding:

> **Finding 2. IDEST can serve as a label-free proxy for SSL pretraining effectiveness:** it decreases as downstream task accuracy improves over the course of unsupervised pre-training.

### 4.3. Label-free metric for Model Selection

As stated in Theorem 3.1 and highlighted by previous experiments, performance on downstream tasks is bounded by the intrinsic dimension of the representations. Consequently, we evaluate **IDEST** as an hyperparameter selection criterion that may bypass the need for linear probing.

**Selecting hyperparameters with IDEST.** Given a set of candidate models $\mathcal{F} = \{f_1, \ldots, f_n\}$, each trained with varying hyperparameters, and their corresponding dimension estimates $\Delta = \{d_1, \ldots, d_n\}$, the selected model, $f^*$, is defined as:

$$f^* = f_{\arg\min_i d_i} \qquad (4)$$

| Dataset | Method | Unsup. | VICReg (RN-50) | | | | DINO (ViT-S) | | | | I-JEPA (ViT-B) | | | |
|---|---|---|---|---|---|---|---|---|---|---|---|---|---|---|
| | | | lr | wd | var. | **all** | lr | s-temp | t-temp | **all** | lr | target-size | context-size | **all** |
| ImageNet | ACC-1 Bounds | | [62.2, 66.5] | [37.5, 69.1] | [65.5, 67.2] | [37.5, 69.1] | [63.6, 69.1] | [48.4,67.8] | [61.2 ,67.5] | [48.4, 69.1] | [61.9, 66.4] | [49.0, 66.4] | [61.3, 66.5] | [49.0,66.5] |
| | $\alpha$-ReQ | ✓ | 65.0 | 53.5 | **66.9** | 53.5 | 63.6 | 58.6 | 61.2 | 58.6 | 61.9 | 49.0 | 61.3 | 49.0 |
| | RankMe | ✓ | **66.5** | **69.1** | 65.5 | **69.1** | 63.6 | **67.8** | 67.5 | 63.6 | 61.9 | 55.9 | 61.3 | 61.9 |
| | LiDAR | ✗ | 65.0 | **69.1** | 66.8 | 65.0 | **68.2** | 65.6 | 65.0 | **65.5** | 63.4 | **66.4** | **66.0** | 63.4 |
| | **IDEST** | ✓ | 62.3 | 67.1 | 65.5 | 65.5 | 64.7 | 65.6 | **67.5** | **65.5** | **66.4** | **66.4** | **66.0** | **66.4** |
| Fine-Grained | ImageNet Oracle | ✗ | 67.2 | 62.2 | 68.6 | 62.9 | 65.5 | 64.8 | 55.2 | 65.5 | 60.0 | 60.7 | 58.3 | 60.0 |
| | $\alpha$-ReQ | ✓ | 66.7 | 47.8 | **68.5** | 66.5 | 63.8 | 59.4 | 60.6 | 59.4 | 59.5 | 41.4 | 57.9 | 41.4 |
| | RankMe | ✓ | **67.2** | **62.2** | 64.9 | 62.9 | 63.8 | **64.8** | 64.5 | **63.8** | 59.5 | 56.9 | 57.9 | 59.5 |
| | LiDAR | ✗ | 66.7 | **62.2** | 66.6 | **66.6** | 68.3 | 62.4 | 64.3 | 62.4 | 58.3 | 60.7 | 58.2 | 58.3 |
| | **IDEST** | ✓ | 61.8 | 58.3 | 64.9 | 65.0 | **63.9** | 62.4 | **64.5** | 62.4 | **60.0** | **60.0** | 58.3 | **60.0** |

*Table 1.* **Unsupervised model selection with IDEST.** We evaluate IDEST for hyperparameter selection against a supervised linear probe on ImageNet-1k, two unsupervised baselines: $\alpha$-ReQ (Agrawal et al., 2022) and RankMe (Garrido et al., 2023), and a weakly-supervised one: LiDAR (Thilak et al., 2024). For each SSL objective (and architecture), hyperparameters are jointly selected across according to Equation (4). 'Fine-grained' denotes the average performance across all furhter datasets (i.e. iNat-21,SUN, Aircraft, CUB, CIFAR-10) excluding ImageNet-1k. **Bold** values indicate the top-performing model selected by the criteria. IDEST is competitive with unsupervised and weakly-supervised baselines across SSL methods, including VICReg, despite RankMe directly mirroring its objective.

**Set-Up.** We apply Equation (4) to identify the optimal model for a given hyperparameter configuration across several SSL frameworks including Joint-Embedding methods (VICReg (Bardes et al., 2022), DINO (Caron et al., 2021)) and the Joint-Predictive architecture (I-JEPA (Assran et al., 2023)). Training details for each model are provided in Appendix B.3.

We focus on various hyperparameters:

1. Optimization: learning rate (lr) and weight decay (wd);
2. Loss-specific coefficients: variance coefficients in VICReg (var.); the teacher and student temperatures in DINO (t-temp., s-temp.);
3. Size of masking for I-JEPA: target block size (target-size) and context block size (context-size).

We evaluate performance on ImageNet and the average accuracy across several fine-grained classification datasets, comparing models selected by IDEST against those selected by supervised validation accuracy, and include the **all** setting, where selection is performed over the full hyperparameter pool, the more challenging configuration.

**Results.** As shown in Table 1, IDEST can recover most ImageNet oracle performance, achieving results near the upper bound of validation accuracy across various architectures (ResNets and ViTs) and pre-training objectives. Notably, IDEST outperforms $\alpha$-ReQ in most settings without suffering from significant performance drops in the worst-case scenarios. For instance, $\alpha$-ReQ selects the lower-bound accuracy for I-JEPA.

Additionally, while RankMe was originally designed for JE-SSL methods (e.g., VICReg, DINO), and in fact mirrors the VICReg objective by directly rewarding high effective rank, IDEST maintains competitive performance on these

| Model | Architecture | $D$ | Param (M) | Linear Probe (min) | IDEST (min) |
|---|---|---|---|---|---|
| VICReg | ResNet-50 | 2048 | 24 | 64.5 | 2.1 |
| DINOv2 | ViT-S | 384 | 22 | 65.8 | 1.8 |
| | ViT-B | 768 | 86 | 64.5 | 2.1 |
| | ViT-L | 1024 | 303 | 113.3 | 4.9 |
| | ViT-G | 1408 | 1000 | 322.7 | 12.8 |

*Table 2.* **Computational cost.** Wall-clock time (min) on ImageNet-1k to evaluate frozen representations using either linear probing (10 epochs, batch size 1024) or IDEST (single feature-extraction pass followed by $\dim_{\text{MST}}$ computation). Rows vary the backbone architecture, primarily changing the output dimension $D$ and the number of parameters. Results are averages over 3 runs.

models while also generalising to settings where RankMe struggles, such as I-JEPA. IDEST demonstrates versatility to other SSL paradigms. As I-JEPA is not a standard Joint-Embedding Method (making dimensional collapse less of a primary concern), other metrics struggle, whereas IDEST successfully generalizes to this and other SSL paradigms.

Furthermore, LiDAR (Thilak et al., 2024) is a strong baseline when pretraining augmentations are accessible. Nonetheless, across all four models in the **all** setting, IDEST achieves comparable model selection than LiDAR in a strictly unsupervised setting. Additional results on DINO (ResNet-50) are available in Table 4.

> **Finding 3.** IDEST **can serve as an efficient label-free criterion for hyperparameter selection:** it performs comparably to supervised baselines across diverse architectures and SSL objectives.

### 4.4. Computational Cost

We evaluate the computational cost of IDEST and compare it to linear probing. Compared to training a linear probe,

which requires multiple epochs over the training dataset features to obtain reasonable estimates of downstream accuracy, computing **IDEST** only requires feature extraction and a $\dim_{\text{MST}}$ computation. In Table 2, we compare the wall-clock compute time for **IDEST** and standard linear probe training on ImageNet (10 epochs across 2 GPUs), averaged over 3 runs. Computations were performed on H100 GPUs and an Intel Sapphire Rapids 8468 CPU. Across all architectures, **IDEST** is substantially faster than a 10-epoch linear probe ($B = 1024$, on 2 GPUs). Notably, for larger models, the cost is dominated by feature extraction, while the overhead of computing $\dim_{\text{MST}}$ remains minimal across various output dimensions.

### 4.5. Limitations

While versatile, our framework has several limitations.

The formula in Theorem 3.1, provides only a bound on the convergence rate. Therefore, even if two models have similar intrinsic dimensions, their actual convergence rates may potentially differ in practice. Thus, **IDEST** should be viewed more as an indicator of accuracy than as a perfect predictor. This is indeed reflected in Figure 1 for instance, where $\rho \approx 0.8, \tau \approx 0.6$ confirms strong global ranking ability, while acknowledging that the intrinsic dimension might not explain all the variance. In particular, ranking is inherently harder when model accuracies are tightly clustered, as in ImageNet or SUN397, than when they are more spread out, as in iNaturalist. Furthermore, one family deviates most visibly: vision-language models (e.g., CLIP, EVA-CLIP), where the cone effect introduces geometric misalignment between encoders that the contrastive objective preserves rather than resolves, constraining the representations (Liang et al., 2022).

Additionally, a more subtle limitation stems from the early-training regime of ViTs: **IDEST** is less informative during the first 10 training epochs, before representations develop stable geometric structure (Section 4.2).

### 5. Conclusion

In this work, we introduced **IDEST**, an unsupervised criterion for evaluating self-supervised representations based on the estimation of the intrinsic dimension via minimum spanning trees ($\dim_{\text{MST}}$). Building on the theoretical connection between intrinsic dimension and generalization, we demonstrated that **IDEST** serves as a robust and an efficient geometric proxy for downstream performance in SSL.

Our empirical evaluation across diverse datasets, architectures, and SSL objectives shows that **IDEST** correlates with supervised linear probing accuracy. Furthermore, **IDEST** provides a principled label-free metric for hyperparameter selection, performing on par with supervised oracles and

generalizing across heterogeneous SSL paradigms.

By offering a unified framework for assessing SSL representations without requiring annotated data, our work highlights the potential of intrinsic geometric descriptors to complement standard evaluation protocols.

**Future Work.** Future work could explore several directions. First, the relationship between intrinsic and effective dimensions (e.g., RankMe (Garrido et al., 2023)) warrants closer study. As further discussed in Section F, a large gap between the two may signal curvature in the representation manifold, opening the door to differential-geometric tools for complementary analysis. Second, integrating intrinsic dimension estimates directly into SSL training objectives could enable geometry-aware learning, guiding models toward representations that are simultaneously compact and well-spread. Finally, extending **IDEST** to dense tasks, e.g., segmentation or generation, is a natural next step, as the underlying MST estimator is task-agnostic, though such settings do not fall within the theoretical framing leveraged here.

## Acknowledgments

This work is supported by Hi! PARIS, ANR/France 2030 program (ANR-23-IACL-0005) and Inria Action Exploratoire PREMEDIT (Precision Medicine using Topology). We were granted access to the HPC resources of IDRIS under the allocations 2025-A0190616899 made by GENCI. We would like to thank David Loiseaux and Eleftherios Tsonis for their useful feedback.

## Impact Statement

This paper presents work whose goal is to advance the field of machine learning. There are many potential societal consequences of our work, none of which we feel must be specifically highlighted here.

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

**Appendix to**

# IdEst: Assessing Self-Supervised Learning Representations via Intrinsic Dimension

## Contents

## A. IdEst's Implementation Details

**Minimum Spanning Tree.**  We complement the definition of the Minimum Spanning Tree (MST) given in Section 3.2 with a visual overview in Figure 6 and with formal definitions below.

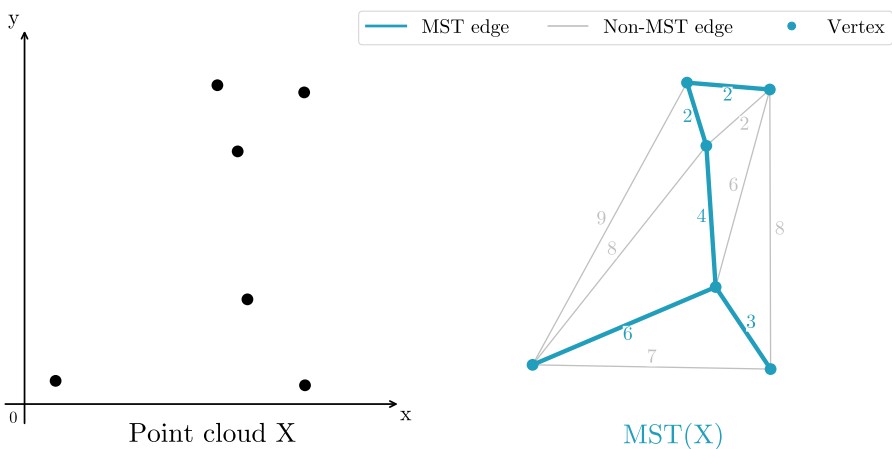

*Figure 6.* **Overview of a Minimum Spanning Tree** (MST). *(Left)* A point cloud $X \subset \mathbb{R}^2$. *(Right)* The MST($X$) connects all points without forming any cycle, while minimizing the total edge length. *Grey edges* indicate pairwise connections not retained in the MST: including them would either create a cycle or increase the total length.

**Definition A.1.** (Spanning Tree). A spanning tree of $X$ is an undirected graph $G = (V, E)$ with vertex set $V = X$ and edge set $E \subseteq V \times V$ such that $G$ is connected and acyclic.

**Definition A.2.** (Minimum Spanning Tree). A minimum spanning tree (MST) of $X$ is a spanning tree $G = (V, E)$ of minimum total edge weight:

$$L\left(\mathrm{MST}(X)\right) := \sum_{(u,v) \in E^*} \|u - v\|_2.$$

$\dim_{\mathbf{MST}}$. As described in Section 3.2, **IDEST** leverages $\dim_{\mathrm{MST}}$ to estimate the intrinsic dimension of the representation space. In practice, the ID is estimated by log-log linear regression over subsamples of increasing size: given subsamples $Z_{n_i}$ with sizes $n_i$, we fit $\log(L\left(\mathrm{MST}(Z_{n_i})\right)) \approx \frac{d-1}{d}\log(n_i) + \log(C)$, and recover $d = 1/(1-m)$ from the fitted slope $m$. The complete algorithm is given in Algorithm 1.

---

**Algorithm 1** Computation of $\dim_{\mathrm{MST}}$

---

    **Input:** The set of representations $Z = \{z_1, \ldots, z_N\}$, minimum sample size $n_{\min}$, skip size $\Delta$
    **Output:** The estimated dimension: $\dim_{\mathrm{MST}}(Z)$
    Initialize $n \leftarrow n_{\min}$, $E \leftarrow []$
    **while** $n < N$ **do**
        $Z_n \leftarrow \mathrm{sample}(Z, n)$  # random sampling of $n$ points
        $E[i] \leftarrow L\left(\mathrm{MST}(Z_n)\right)$  # computation of the minimum spanning tree
        $n \leftarrow n + \Delta$
    **end while**
    $m, b \leftarrow$ linear regression$(\log(n_{\min} : \Delta : N), \log(E))$  # linear regression of the log-log plot
    $\dim_{\mathrm{MST}}(\mathrm{Z}) \leftarrow 1/(1-m)$

---

We follow the implementation of Adams et al. (2020); Dupuis et al. (2023) for $\dim_{\mathrm{MST}}$. For each dataset and model, we compute the dimension estimator. Preprocessing (distance matrix computation and sorting) dominates in practice and is optimized via Ripser (Tralie et al., 2018; Bauer, 2021), which implements an efficient sparse distance pipeline. To keep computation tractable, **IDEST** operates on a subsample of size $N \ll N_D$, where $N_D$ denotes the full dataset size; we set $N = 50{,}000$ throughout.

## B. Implementation Details for SSL models studied

### B.1. Image Classification Training Details

**Datasets.** We evaluate the global quality of the SSL models using the widely adopted linear probing evaluation. We consider:

1. ImageNet dataset (Deng et al., 2009)
2. Large-scale fine-grained datasets: iNat-18 (Van Horn et al., 2018), iNat-21 (Van Horn et al., 2021), SUN397 (Xiao et al., 2010)
3. Small-scale fine-grained datasets: CIFAR-10 and CIFAR-100 (Krizhevsky et al., 2009), Aircraft (Maji et al., 2013), CUB200 (Welinder et al., 2010)

**Evaluation protocol.** For each baseline, we follow the protocol of (Siméoni et al., 2025) and train a linear layer on the final frozen representation. To obtain the frozen representation, we follow each method's standard evaluation protocol, e.g., the `CLS` token after the layer norm, or `avgpool` if there is no `CLS` token. Specifically, we use SGD with a momentum of 0.9, and train for 10 epochs with a batch size of 1024, using random-resized-crop data. We perform the following grid search:

- **Learning Rate** in $\{0.0001, 0.0002, 0.0005, 0.001, 0.002, 0.005, 0.01, 0.02, 0.05, 0.1, 0.2, 0.3, 0.5, 1\}$

For the fine-grained dataset (i.e., Aircraft), following (Oquab et al., 2023), we use a lighter weight evaluation using scikit-learn's LogisticRegression implementation with the L-BFGS solver.

| Method | Architecture | Repo | Top-1 Accuracy (%) | | | | |
|---|---|---|---|---|---|---|---|
| | | | ImageNet | v2 | SUN397 | iNat-18 | iNat-21 |
| BarlowTwins (Zbontar et al., 2021) | ResNet-50 | github | 72.9 | 60.3 | 71.6 | 38.1 | 53.4 |
| CLIP (Radford et al., 2021) | ResNet-50 | timm | 69.6 | 52.8 | 76.5 | 13.7 | 46.4 |
| | ViT-B/16 | timm | 80.1 | 68.8 | 82.2 | 51.1 | 63.4 |
| | ViT-B/32 | timm | 76.1 | 63.5 | 80.6 | 41.9 | 54.6 |
| | ViT-L/14 | timm | 84.3 | 75.1 | 84.8 | 59.8 | 72.5 |
| DINO (Caron et al., 2021) | ViT-S/16 | github | 76.0 | 64.4 | 72.5 | 48.4 | 60.9 |
| | ViT-B/16 | github | 77.8 | 66.7 | 74.8 | 49.9 | 63.4 |
| DINOv2 (Oquab et al., 2023) | ViT-S/14 | github | 80.4 | 70.4 | 79.5 | 61.5 | 73.5 |
| | ViT-B/14 | github | 83.8 | 74.9 | 81.7 | 68.3 | 80.3 |
| | ViT-B/14-reg | github | 83.9 | 75.1 | 81.9 | 67.4 | 79.4 |
| | ViT-L/14 | github | 85.7 | 77.7 | 82.8 | 71.3 | 83.1 |
| | ViT-L/14-reg | github | 86.1 | 77.9 | 83.3 | 71.0 | 83.3 |
| | ViT-G/14 | github | 86.2 | 78.1 | 82.9 | 74.0 | 85.3 |
| DINOv3 (Siméoni et al., 2025) | ViT-S/16 | github | 79.3 | 68.7 | 79.1 | 61.8 | 69.4 |
| | ViT-B/14 | github | 84.5 | 75.0 | 82.9 | 70.6 | 79.4 |
| | ViT-L/16 | github | 87.2 | 79.5 | 84.6 | 76.1 | 84.2 |
| | ViT-7B | github | 88.4 | 81.4 | 85.4 | 80.2 | 88.7 |
| EVA (Sun et al., 2023) | EVA01-g-14 | timm | 86.1 | 71.5 | 85.5 | 64.9 | 79.5 |
| | EVA02-b-16 | timm | 83.6 | 73.3 | 83.1 | 55.7 | 70.5 |
| | EVA02-E-14-plus | timm | 87.6 | 75.6 | 86.5 | 70.6 | 82.7 |
| Franca (Venkataramanan et al., 2025) | ViT-L/14 (In-21k) | github | 84.2 | 75.4 | 80.7 | 73.5 | 70.6 |
| iBoT (Zhou et al., 2022) | ViT-S/16 | github | 76.9 | 65.6 | 73.6 | 48.1 | 60.6 |
| | ViT-B/16 | github | 79.2 | 68.1 | 75.2 | 48.6 | 63.4 |
| | ViT-L/16 | github | 80.6 | 69.9 | 76.4 | 51.5 | 66.2 |
| I-JEPA (Assran et al., 2023) | ViT-G/16 | github | 75.8 | 63.8 | 74.9 | 45.9 | 58.7 |
| PE-Core (Bolya et al., 2025) | S16-336 | timm | 70.9 | 51.2 | 75.6 | 38.6 | 54.6 |
| | B14-224 | timm | 83.2 | 72.7 | 83.8 | 60.8 | 74.9 |
| | L14-336 | timm | 87.8 | 78.7 | 87.0 | 74.1 | 84.1 |
| SigLIP (Zhai et al., 2023) | ViT-B-16-SigLIP | timm | 82.5 | 68.9 | 82.7 | 53.3 | 69.6 |
| | ViT-L-16-SigLIP-256 | timm | 86.0 | 73.8 | 85.1 | 59.4 | 70.8 |
| VICReg (Bardes et al., 2022) | ResNet50 | github | 73.0 | 59.9 | 71.7 | 40.2 | 53.8 |

*Table 3.* **Foundation Models Studied.** Overview of the pretrained models evaluated in this work, spanning diverse SSL and vision-language objectives and architectures (ResNet, ViT). Top-1 accuracy is reported on ImageNet, ImageNet-v2, and additional fine-grained datasets SUN397, iNat-18, iNat-21. Model weights are loaded from the official repositories or `timm`.

## B.2. Models studied in Section 4.1

The checkpoints used were either downloaded from the original gihub or timm. The complete list of pretraiend models is reported in Table 3.

## B.3. Models trained in Section 4.3

The complete list of models pretrained can be found below.

**VICReg (Bardes et al., 2022).** VICReg maximizes the informational content of embeddings by regularizing their empirical covariance matrix.

VICReg's loss is defined with three components: *(i)* a term to encourage the variance (diagonal of the covariance matrix) inside the current batch to be equal to 1, preventing collapse with all the inputs mapped on the same vector; *(ii)* and a correlation regularization, encouraging the off-diagonal coefficients of the empirical covariance matrix to be close to 0, decorrelating the different dimensions of the embeddings. *(iii)* an invariance loss that matches positive pairs

We pre-trained ResNet-50 for 100 epochs using LARS, the projector used is an MLP with intermediate dimensions (8192, 8192, 2048), with a batch size of 2048, following the protocol of (Garrido et al., 2023)

1. `lr`: wd $= 1e-6$, lr $\in \{0.1, 0.2, 0.3, 0.4, 0.5\}$, inv : 25, cov : 5, var : 25

2. `wd`: lr $= 0.3$, wd $\in \{1e-7, 1e-6, 1e-5, 1e-4, 1e-3\}$, inv : 25, cov : 5, var : 25

3. `cov.`: lr $= 0.3$, wd $= 1e-6$, inv : 25, cov : 5, var : 25 cov $\in \{0.4, 0.6, 0.8, 1, 4, 16\}$, var : 25

**DINO (Caron et al., 2021).** DINO uses a student-teacher framework. Two versions of the same network (the student and the teacher) are fed different views of the same image. The student is trained to match the teacher's output probability distribution. To prevent the model from collapsing (i.e., giving the same output for every image), DINO uses a unique centering and sharpening operation on the teacher's outputs. The teacher's weights are updated as an exponential moving average of the student's weights.

We pre-trained ViT-S for 100 epochs using Adam-W. The projector used is an MLP with intermediate dimensions (8192, 8192, 32768), with a batch size of 2048:

1. `lr`: lr $\in \{1.25e-4, 2.25e-4, 0.0025, 0.002, 0.0075\}$, t-temp. $= 0.04$, s-temp $= 0.07$

2. `s-temp`: lr $= 0.002$, t-temp. $= 0.04$, s-temp $= \{0.07, 0.1, 0.2, 0.3, 0.4\}$,

3. `t-temp`: lr $= 0.002$, t-temp. $= \{0.01, 0.02, 0.03, 0.04, 0.05\}$, s-temp $= 0.07$,

**I-JEPA (Assran et al., 2023)** I-JEPA is a Joint-Predictive Architecture. It uses a **context block** to predict the representations of several target blocks from the same image. The context encoder is a Vision Transformer (ViT), which only processes the visible context patches. The predictor is a smaller ViT which takes the context encoder output and, conditioned on positional tokens, predicts the representations of a **target block** at a specific location. The weights of the target encoder are updated at each iteration via an exponential moving average of the context encoder weights.

We pre-trained ViT-B for 300 epochs with the same protocol as in the original papers with a batch size of 4096:

1. `lr`: wd sch $= [0.04-0.4]$, lr $\in \{4e-5, 8e-5, 9e-5, 1e-4, 1.25e-4, 2e-4, 3e-4\}$, target block size $= \{0.15, 0.2\}$, context block size $= \{0.85, 1.0\}$

2. `Target Size Block` (the size of the target block): wd sch $= [0.04-0.4]$, lr $= 1.25e-5$, target block size $\in \{\{0.075, 0.2\}, \{0.1, 0.2\}, \{0.125, 0.2\}, \{0.2, 0.25\}, \{0.2, 0.25\}\}$, context block size $= \{0.85, 1.0\}$

3. `Context Size` (the size of the context block): wd sch $= [0.04-0.4]$, lr $= 1.25e-5$, target block size $= \{0.15, 0.2\}$, context block size $\in \{\{0.4, 1.0\}, \{0.5, 1.0\}, \{0.65, 1.0\}, \{0.75, 1.0\}, \{0.90, 1.0\}\}$

## C. Additional hyperparameter selection results.

To further validate **IDEST**'s performance on ResNet, we conducted additional experiments on DINO with a ResNet-50 varying `s-temp` (the student temperature) and the `t-temp` (teacher temperature), and with the `all` column in which methods must select from the full pool of hyperparameter configurations. The results are reported in Table 4

| Method | Unsup. | DINO (ResNet-50) s-temp. | t-temp. | all |
|---|---|---|---|---|
| ACC-1 Bounds | | [57.9, 67.5] | [63.0, 68.4] | [57.9, 68.4] |
| $\alpha$-ReQ | ✓ | 61.9 | 63.0 | 63.0 |
| RankMe | ✓ | 61.9 | 67.3 | **67.3** |
| LiDAR | ✗ | 65.5 | **67.3** | 67.3 |
| **IDEST** | ✓ | **67.5** | **67.6** | **67.6** |

*Table 4.* **Unsupervised model selection with IDEST for DINO with a ResNet-50 backbone.** We evaluate **IDEST** for hyperparameter selection against a supervised linear probe on ImageNet-1k, two unsupervised baselines: $\alpha$-ReQ (Agrawal et al., 2022) and RankMe (Garrido et al., 2023), and a weakly-supervised one: LiDAR (Thilak et al., 2024). Hyperparameters are jointly selected across all hyperparameter axes according to Equation (4). **Bold** values indicate the top-performing model selected by the criteria.

## D. Compute cost

All trainings were performed on H100 GPUs. The total computational cost of the project, including training baselines, experiments, and ablation studies, amounts to approximately 12,000 GPU-hours.

## E. Further study of Foundation Models

### E.1. Additional properties studied in Joint-Embedding Self-Supervised Learning

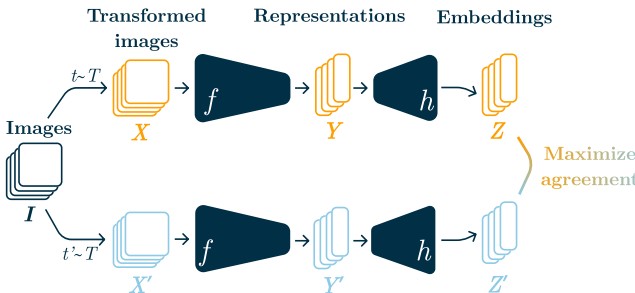

*Figure 7.* **Overview of Joint-Embedding Architectures.** Two separate data augmentation operators are sampled from the same distribution ($t, t' \sim \mathcal{T}$) and applied to each image in a batch $I$ to obtain two views, $X$ and $X'$. An encoder network $f$ and a projection head $g$ are trained to maximize agreement between the resulting embeddings. After training, the encoder $f$ and its representations $Y$ are retained for downstream tasks.

A dominant approach in SSL is *joint embedding self-supervised learning* (JE-SSL) (Chen et al., 2020; Bardes et al., 2022), where two networks are trained to produce similar embeddings for different views of the same image. An overview is presented in Figure 7. Several metrics have been proposed to assess the quality of the learned representations in these methods.

**Uniformity metrics.** Uniformity is a desirable properties of Joint-Embedding Self-Supervised methods (Wang & Isola, 2020; Fang et al., 2024; Mordacq et al., 2025), with the intuition that vectors should be roughly uniformly distributed on the unit hypersphere $S^{m-1}$, preserving as much information of the data as possible. Two main metrics have been proposed:

- $\mathcal{L}_u$ (Wang & Isola, 2020), based on the gaussian pairwise kernel:

$$\mathcal{L}_u = \log \mathop{\mathbb{E}}_{\substack{\text{i.i.d.} \\ x, y \sim p_{\text{data}}}} \exp^{-t||f(x)-f(y)||_2^2}, \ t > 0. \tag{5}$$

- $\mathcal{W}_2$ (Fang et al., 2024), the quadractic Wasserstein distance between the distribution of the learned representation and $\mathcal{N}(0, I_M/m)$:

$$\mathcal{W}_2 := \sqrt{||\hat{\mu}||_2^2 + 1 + \text{tr}(\hat{\Sigma}) - \frac{2}{m}\text{tr}(\hat{\Sigma}^{\frac{1}{2}})} \tag{6}$$

where $\hat{\mu}, \hat{\Sigma}$ are the sample mean and covariance mean.

Though neither was designed with model comparison or hyperparameter selection in mind, we nonetheless investigate their potential across SSL paradigms.

**RankMe (Garrido et al., 2023).** RankMe is formally the smooth rank measure, originally introduced by Roy & Vetterli (2007):

$$\text{RankMe}(Z) = \exp\left(-\sum_{k=1}^{\min(N,K)} p_k \log p_k\right), \text{with } p_k = \frac{\sigma_k(Z)}{||\sigma(Z)||_1} + \epsilon \tag{7}$$

where $Z$ is the representations obtained.

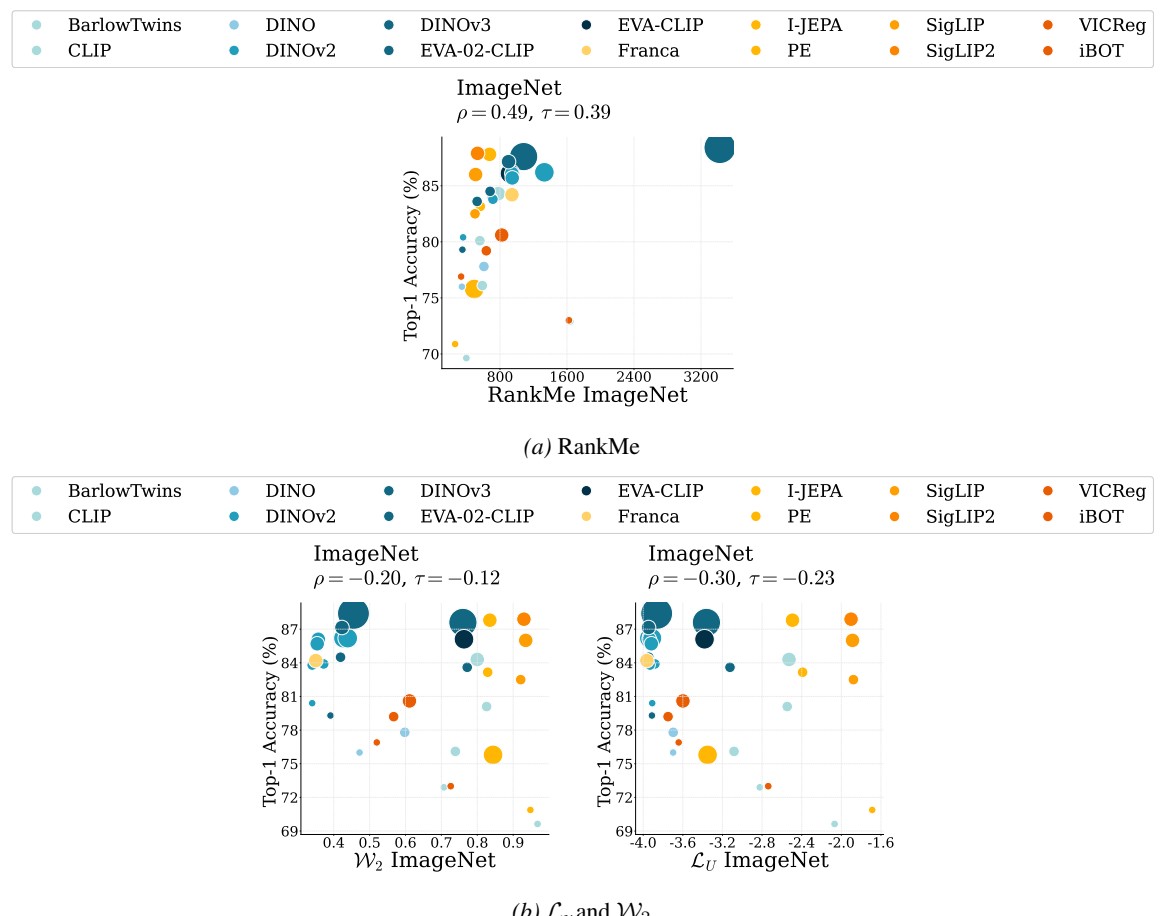

*(a)* RankMe

*(b)* $\mathcal{L}_u$ and $\mathcal{W}_2$

*Figure 8.* **Alternative metrics and Foundation Models.** Linear probing accuracy on ImageNet versus *(Top)* RankMe and *(Bottom)* uniformity metrics ($\mathcal{L}_u$ and $\mathcal{W}_2$) for a diverse set of pretrained SSL models.

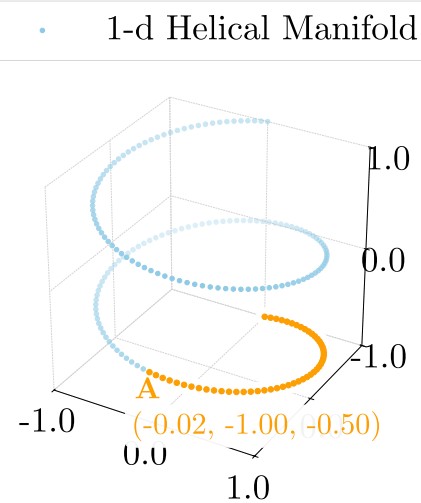

*Figure 9.* **Effective vs. intrinsic dimensionality.** A helix is embedded in a 3-dimensional ambient space (high effective dimension), yet is intrinsically a 1-dimensional manifold: point A requires three coordinates to specify its position in the ambient space, but a single arc-length coordinate suffices to locate it along the curve. The two notions thus capture complementary aspects of geometry—how spread out a representation is versus how many degrees of freedom it truly contains.

### E.2. Foundation Models and further metrics.

In Figure 8 we study whether metrics initially proposed for JE-SSL can reflects representation quality across a diverse set of pretrained SSL models. Specifically, we compute RankMe, $\mathcal{L}_u$, $\mathcal{W}_2$ on frozen representations and compare it against standard linear probing accuracy on ImageNet.

For RankMe (Garrido et al., 2023), the original study noted that it is not suited for comparisons across architectures. A key reason is that RankMe is bounded by the output dimension of the model: two models of different architectures that both span their full representation space will yield different values, making cross-architecture comparisons unreliable (as highlighted in Figure 8a, where DINOv3 ViT-7B is a clear outlier compared to DINOv3 ViT-L, despite similar ImageNet accuracies).

Concerning uniformity metrics (see Figure 8b), neither $\mathcal{L}_u$ nor $\mathcal{W}_2$ correlates consistently with linear probing accuracy across models.

Overall, all three metrics retain some predictive signal, as reflected by non-trivial Kendall's $\tau$ and Spearman's $\rho$ values, yet their limited correlation with accuracy motivates the search of alternative geometric proxies.

## F. Effective and Intrinsic Dimensions

RankMe (Garrido et al., 2023) and **IDEST** capture complementary notions of dimensionality. The former measures *effective rank*, the entropy of the singular-value distribution of the feature matrix, reflecting how uniformly the representation spreads across linear dimensions (Roy & Vetterli, 2007). The latter estimates *intrinsic dimension*: the minimum number of degrees of freedom required to describe the data on its underlying manifold. Figure 9 illustrates this distinction with a helix, whose points require three ambient coordinates yet lie on a curve parameterised by a single value.

RankMe correlates less strongly with downstream accuracy across foundation models (Figure 8; $\rho = 0.49$, $\tau = 0.39$) than **IDEST** (Figure 1, $\rho = -0.74$, $\tau = -0.55$), confirming that intrinsic dimension captures information about representation quality beyond what linear spread alone can reveal. Yet RankMe's substantial correlation suggests that the effective rank of representations remains a useful proxy for downstream performance.

Figure 10 tracks the training dynamics of DINO (ViT-S) and reveals a consistent trend: **IDEST** decreases while RankMe increases throughout training. This inverse relationship admits a natural information-bottleneck interpretation: high-quality representations *compress* the input onto a compact, low-dimensional manifold (low intrinsic dimension) while *spreading* that information uniformly across ambient dimensions (high effective rank) to avoid collapse.

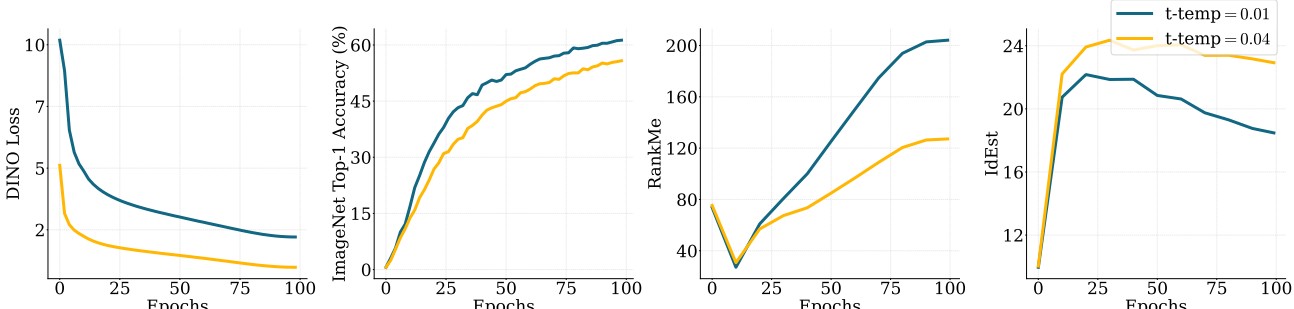

*Figure 10.* **Tracking Training Dynamics: RankMe and IDEST.** Evolution of the self-supervised loss, ImageNet-1k online classification top-1 accuracy, **IDEST** and RankMe. **IDEST** decreases while RankMe increases throughout training.

As hypothesized in (Ansuini et al., 2019), the gap between effective and intrinsic dimension relates to the *curvature* of the representation manifold: a flat manifold embedded in $\mathbb{R}^d$ has matching intrinsic and effective dimensions, whereas a highly curved one can occupy many ambient dimensions while remaining intrinsically low-dimensional, as illustrated by the helix in Figure 9. Differential geometry offers a principled framework to formalise this gap; curvature-aware metrics are a natural direction for future work to further disentangle the structure of learned representations.

