# OpenReview forum: "IdEst: Assessing Self-Supervised Learning Representations via Intrinsic Dimension"
_ICML.cc/2026/Conference — ICML 2026 regular_

### Official Review · Reviewer_R6oR · 2026-03-08

**Soundness:** 3
**Presentation:** 3
**Significance:** 4
**Originality:** 4
**Overall Recommendation:** 4
**Confidence:** 4

**Summary:**

The manuscript introduces IDEST, a novel, label-free method designed to estimate the intrinsic dimension of representations generated by Self-Supervised Learning models. The authors argue that the standard evaluation protocol, linear probing, is computationally demanding and fails to provide insights into the underlying geometric structure of the feature space. To address this, IDEST leverages the length of Minimum Spanning Trees to estimate intrinsic dimension, offering a global geometric perspective that is more robust than traditional local parametric estimators. Through extensive empirical testing across ten different self-supervised frameworks and multiple datasets, the paper demonstrates that IDEST strongly correlates with downstream classification accuracy. Furthermore, the authors show that this metric can effectively track training dynamics and serve as a reliable unsupervised criterion for hyperparameter selection.

**Compliance With Llm Reviewing Policy:**

Affirmed.

**Final Justification:**

I thank the authors for their detailed and constructive rebuttal. After reading the response and the other reviews, my final recommendation remains a Weak Accept. The rebuttal successfully reinforced my prior positive assessment, though it did not warrant an increase in the final score.

**Key Questions For Authors:**

1. As mentioned in the first weakness, a comparison of the time overhead of computing IDEST versus training a linear probe should be provided.
2. It should be discussed whether IDEST's ability to track downstream performance can be generalized to CNN architectures like ResNet.
3. It should be discussed under which specific dataset distributions, SSL paradigms, or extreme crash scenarios IDEST might fail; this remains unclear.

**Limitations:**

yes

**Strengths And Weaknesses:**

Strengths
1. The paper tackles a significant bottleneck in self-supervised learning research. Developing robust, unsupervised metrics to evaluate representation quality without the computational overhead of training linear probes is valuable to the community.
2. The experimental setup is impressively thorough. The inclusion of diverse architectures (ResNets and Vision Transformers), various pretraining paradigms (joint-embedding, masked image modeling, vision-language), and multiple fine-grained datasets provides strong evidence for the generalizability of IDEST.

Weaknesses
1. Lack of Wall-clock Time Comparison.
While the authors assert that IDEST significantly reduces the computational burden compared to supervised evaluation protocols, the manuscript lacks a direct wall-clock time comparison between computing IDEST and training a linear probe. Without this empirical runtime evidence, the core motivation regarding computational efficiency remains unsubstantiated.
2. Limited Architecture Diversity in Online Probing.
Although the paper utilizes a diverse set of architectures for offline evaluation , the online probing experiments used to track training dynamics (Section 4.2.2) are validated solely on a single model, DINO with a ViT-S backbone. It remains unclear whether IDEST's capacity to smoothly track downstream performance can generalize to CNN architectures like ResNet.
3. Absence of Failure Mode Analysis.
Although Figure 1 shows a general correlation, several data points visibly deviate from the trend. The paper lacks a deeper investigation into these anomalies, leaving it unclear under which specific dataset distributions, SSL paradigms, or extreme collapse scenarios IDEST might fail.

---

> ### Author Rebuttal · Authors · 2026-03-30
>
> We thank you for your feedback and positive comments. Please find below our answers to the weaknesses you raised:
>
> **W1 & Q1  “Lack of Wall-clock Time Comparison. [...]” / “As mentioned in the first weakness, a comparison of the time overhead of computing IDEST [...].”**
>
>
> $\mathrm{MST}$s are computed via Kruskal's algorithm [1] with worst-case complexity $\mathcal{O}(N^2 \cdot (D + \log N))$, where $N$ is the number of samples and $D$ is the embedding dimension. Preprocessing (distance matrix computation and sorting) dominates in practice and is optimized via Ripser [2], which implements an efficient sparse distance pipeline.
>
> IdEst can operate on a subsample of size $N \ll N_D$, where $N_D$ is the full dataset size; sensitivity analyses over 30 models (same as in Fig. 1; Appendix B.2) show that at $N = 50,000$, Spearman rank correlation reaches $\rho=0.99$, indicating that relative model rankings are highly stable, with only a 3.2% average change in estimated values, justifying this choice.
>
>
> As for the practical runtime, IdEst requires only a single feature-extraction pass followed by the $\mathrm{dim}_\mathrm{MST}$ computation. Table B.1 compares wall-clock time against a 10-epoch linear probe  under a single hyperparameter configuration ($B=1024$, batch size; 2 H100 GPUs) on ImageNet-1k (avg. over 3 runs):
>
>
> #### Table B.1. Computational cost.
>
> | Model | Architecture |  D | Param (M) | Linear Probe (min) | IdEst (min) |
> |:-|:-|:-:|:-:|:-:|:-:|
> | VICReg | ResNet-50 | 2048 | 24 | 64.5 | 2.3 |
> | DINOv2 | ViT-S | 384 | 22 | 65.8 | 1.8 |
> | | ViT-B | 768 | 86 | 64.5 | 2.1 |
> | | ViT-L | 1024 | 303 | 113.3 | 4.9 |
>
> IdEst reduces evaluation time by ${\sim}26{\times}$ on average compared to linear probing (which may require additional hyperparameter tuning, e.g., learning rate selection).
>
> **W2 & Q2. Limited Architecture Diversity in Online Probing. Although the paper utilizes a diverse set of architectures for offline evaluation, the online probing experiments used to track training dynamics (Section 4.2.2) are validated solely on a single model, DINO with a ViT-S backbone. [...]" / "It should be discussed whether IDEST's ability to track downstream performance can be generalized to CNN architectures like ResNet."**
>
>
> IdEst can indeed generalize to ResNet architectures. Table D.2 reports IdEst alongside online validation accuracy during VICReg training with a ResNet-50 backbone on ImageNet-1k (100 epochs).
>
> #### Table D.2 Online probing dynamics during VICReg training with a RN-50
>
> | Epochs | VICReg Loss |Top-1 Acc (%) | IdEst |
> |-|:-:|:-:|:-:|
> | 1 | 39.2| 1.1 | 79.9 |
> | 20 | 24.6 | 54.1 | 25.7 |
> | 40 | 22.8 | 60.4 |  23.8 |
> | 60 | 21.3 | 64.1 |  22.4 |
> | 80 | 20.1 | 66.4 |  21.6 |
> | 100 | 19.7 | 66.9 |  20.3 |
>
> IdEst decreases throughout training, closely mirroring the rise in validation accuracy, consistent with our DINO/ViT-S observations (Section 4.2.2). It confirms its reliability across backbone architectures and training paradigms.
>
>
> **W3 & Q2 "Absence of Failure Mode Analysis. Although Figure 1 shows a general correlation, several data points visibly deviate from the trend. [...]" / Q2. "It should be discussed under which specific dataset distributions, SSL paradigms [...]"**
>
> The formula in Theorem 3.1, $\mathcal{L} \approx \mathcal{O}(K_L \cdot N^{-\frac{1}{d}})$, provides only a bound on the convergence rate. Therefore, even if two models have similar intrinsic dimensions, their actual convergence rates may potentially differ in practice.  Thus, IdEst should be viewed more as an indicator of accuracy than as a perfect predictor. This is indeed reflected in Figure 1, where $\rho \approx -0.8$ confirms strong global ranking ability, while $R^2 \approx 0.6$  acknowledges that the intrinsic dimension $d$ might not explain all the variance.
>
> Two families deviate most visibly: redundancy-reduction methods (VICReg, BarlowTwins), whose objectives explicitly aim to increase dimensionality by design; and vision-language models (e.g., CLIP, EVA-CLIP), where the cone effect introduces geometric misalignment between encoders that the contrastive objective preserves rather than resolves, constraining the representations [3].
>
> Additionally, a more subtle limitation stems from the early-training regime of ViTs: IdEst is less informative during the first ~20 training epochs of ViTs, before representations develop stable geometric structure (Section 4.2).
>
>
> [1] J.B. Kruskal, On the shortest spanning subtree of a graph and the traveling salesman problem. In Proceedings of the American Mathematical society, 1956
>
> [2] Tralie et al, Ripser. py: A lean persistent homology library for python, JOSS 2018
>
> [3] Mind the gap: Understanding the modality gap in multi-modal contrastive representation learning, Liang et al, NeurIPS 2022

---

> > ### Author Rebuttal · Reviewer_R6oR · 2026-04-02
> >
> > I have no further questions. Thank you for your hard work.

---

### Official Review · Reviewer_FZxs · 2026-03-12

**Soundness:** 3
**Presentation:** 3
**Significance:** 3
**Originality:** 1
**Overall Recommendation:** 3
**Confidence:** 5

**Summary:**

This paper proposes ID-Est, an unsupervised metric for evaluating self-supervised learning representations based on estimating their intrinsic dimension (ID) using a minimum spanning tree–based estimator (dimMST). Motivated by theoretical connections between intrinsic dimension and model generalization, the authors assume that high-quality representations lie on lower-dimensional manifolds. They empirically demonstrate that ID-Est correlates strongly with downstream linear probing accuracy across multiple SSL paradigms, architectures, and datasets. The method is further shown to track representation quality during training and to enable efficient label-free hyperparameter selection, reducing the need for computationally expensive supervised evaluation.

**Compliance With Llm Reviewing Policy:**

Affirmed.

**Key Questions For Authors:**

Scalability:
What is the computational complexity of ID-Est when applied to very large datasets or extremely high-dimensional embeddings typical of modern applications?

Generality:
The experiments focus primarily on vision SSL models. Do the authors expect ID-Est to behave similarly for different embeddings and representations?

**Limitations:**

The limitations section does not sufficiently discuss practical constraints such as computational scalability, potential failure cases where intrinsic dimension may not correlate with downstream performance, or the implications of relying on geometric proxies for representation quality.

**Strengths And Weaknesses:**

The paper builds on established theory linking intrinsic dimension to generalization and using a principled estimator (dimMST) with known robustness properties. The empirical evaluation is fairly extensive, covering multiple SSL paradigms, architectures (ResNet and ViT), and datasets. The presentation is generally clear, with well-motivated theoretical background and structured experimental sections.

However, the core idea (using intrinsic dimension as a proxy for representation quality) is not entirely new, and the primary novelty lies in the choice of estimator and its application to SSL evaluation.

P.S.: Some presentation choices in the paper such as the "Takeaways" are not appropriate in a research paper.

---

> ### Author Rebuttal · Authors · 2026-03-30
>
> We thank the reviewer for their feedback. Please find below our answers to the weaknesses and questions you raised:
>
> **W1. “However, the core idea (using intrinsic dimension as a proxy for representation quality) is not entirely new, and the primary novelty lies in the choice of estimator and its application to SSL evaluation.”**
>
>
> All methods presented in this paper for assessing vision SSL (e.g., RankMe, $\alpha$-ReQ) aim to uncover structural properties of learned data representations in an unsupervised manner, in order to predict performance on downstream tasks. Intrinsic dimension estimation is one way to do so, and previous methods such as RankMe also use well-established dimension estimators such as the effective rank [1] for this purpose.
>
>
> The use of dimension estimation is challenging in the context of vision SSL as it requires to work under assumptions far from the usual asymptotic conditions: $n \approx d$ instead of $n \longrightarrow  \infty$ for constant $d$.  As a result, many classic estimators fail: for instance, as reported in (Fig. 6) [2], TwoNN is unable to properly detect the intrinsic dimension and even diverges. In this context, finding a good estimator is a valuable contribution.
>
>
> Our experiments show that MST-based estimation is robust to sampling sparsity, a behavior connected to known theoretical results on MST length in sparse regimes (e.g., Theorem 1 in [3]), enabling IdEst to reliably track representation quality across SSL paradigms (e.g., Figure 1).
>
>
> We intend to add a few sentences along these lines to clarify our contribution in the introduction and Section 3.
>
>
> **"Takeaways are not appropriate in a research paper."**
> The boxes were intended to improve readability and link each subsection's conclusion to the next. We will modify their layout and titles accordingly.
>
> **Q1. "Scalability: What is the computational complexity of ID-Est when applied to very large datasets or extremely high-dimensional embeddings typical of modern applications?"**
>
> $\mathrm{MST}$s are computed via Kruskal's algorithm [1] with worst-case complexity $\mathcal{O}(N^2 \cdot (D + \log N))$, where $N$ is the number of samples and $D$ is the embedding dimension. Preprocessing (distance matrix computation and sorting) dominates in practice and is optimized via Ripser [2], which implements an efficient sparse distance pipeline.
>
> IdEst can operate on a subsample of size $N \ll N_D$, where $N_D$ is the full dataset size; sensitivity analyses over 30 models (same as in Fig. 1; Appendix B.2) show that at $N = 50,000$, Spearman rank correlation reaches $\rho=0.99$, indicating that relative model rankings are highly stable, with only a 3.2% average change in estimated values, justifying this choice.
>
>
> As for the practical runtime, IdEst requires only a single feature-extraction pass followed by the $\mathrm{dim}_\mathrm{MST}$ computation. Table B.1 compares wall-clock time against a 10-epoch linear probe  under a single hyperparameter configuration ($B=1024$, batch size; 2 H100 GPUs) on ImageNet-1k (avg. over 3 runs):
>
>
> #### Table B.1. Computational cost.
>
> | Model | Architecture |  D | Param (M) | Linear Probe (min) | IdEst (min) |
> |:-|:-|:-:|:-:|:-:|:-:|
> | VICReg | ResNet-50 | 2048 | 24 | 64.5 | 2.3 |
> | DINOv2 | ViT-S | 384 | 22 | 65.8 | 1.8 |
> | | ViT-B | 768 | 86 | 64.5 | 2.1 |
> | | ViT-L | 1024 | 303 | 113.3 | 4.9 |
>
> IdEst reduces evaluation time by ${\sim}26{\times}$ on average compared to linear probing (which may require additional hyperparameter tuning, e.g., learning rate selection).
>
>
> **Q2. "Generality: The experiments focus primarily on vision SSL models. Do the authors expect ID-Est to behave similarly for different embeddings and representations?"**
>
> We are confident that IdEst generalizes beyond vision SSL. Both Theorem 3.1 and $\mathrm{dim}_\mathrm{MST}$ are modality-agnostic, requiring only that representations concentrate near a lower-dimensional manifold, a hypothesis well-supported across modalities [4]. Our related work section further surveys intrinsic dimension estimates for LLM representations, including AI-generated text detection [5], supporting broader applicability.
>
>
> **Limitations** Please refer to Q1 for scalability, and to W3 of R-R6oR for additional limitations.
>
> [1] Roy et al, The effective rank: A measure of effective dimensionality, 2007
>
> [2] Birdal et al, Intrinsic Dimension, Persistent Homology and Generalization in Neural Networks, NeurIPS 2024
>
> [3] Mordacq et al, T-REGS: Minimum Spanning Tree Regularization for Self-Supervised Learning, NeurIPS 2025
>
> [4] Goodfellow, et al. Deep learning, MIT press, 2016
>
> [5] Tulchinskii et al, Itrinsic Dimension Estimation for Robust Detection of AI-Generated Texts, NeurIPS 2023

---

> > ### Author Rebuttal · Reviewer_FZxs · 2026-04-04
> >
> > W1 (Novelty of intrinsic dimension as a proxy):
> > The authors clarify that the contribution lies in identifying a robust estimator (MST-based) suitable for SSL settings and explain why existing estimators fail in non-asymptotic regimes. This significantly improves the positioning of the work while also confirming that the core idea itself is not new.
> >
> > W1 (Takeaways):
> > The authors acknowledge the issue and commit to revising the presentation. This point is fully resolved.
> >
> > Q1 (Scalability):
> > This concern is well addressed. The authors provide complexity analysis, practical implementation details, and runtime comparisons.
> >
> > Q2 (Generality):
> > The authors argue that the method is modality-agnostic and support this with theoretical reasoning and references to prior work in other domains. While this is reasonable, the evidence remains indirect since experiments are limited to vision SSL. This concern is partially resolved.

---

> > > ### Author Response · Authors · 2026-04-07
> > >
> > > We thank the reviewer for their constructive feedback.
> > >
> > > Regarding Q2, extending our approach to other modalities is indeed a compelling direction, but a thorough investigation will require significant additional experimentation. As mentioned at the end of the conclusion of the paper, we leave this as future work.

---

### Official Review · Reviewer_B9sk · 2026-03-13

**Soundness:** 3
**Presentation:** 2
**Significance:** 3
**Originality:** 3
**Overall Recommendation:** 4
**Confidence:** 2

**Summary:**

In order to provide more insights into the geometric structure of the learned representation space from self-supervised learning, this work propose a method IDEST to estimate the intrinsic dimensionality of self-supervised learning representation via the Minimum Spanning Tree dimension estimator. The proposed method exhibits high performance across different datasets, architetures and learning objectives. Also, it is more cost-efficient compared to supervised methods.

**Compliance With Llm Reviewing Policy:**

Affirmed.

**Key Questions For Authors:**

See Weakness

**Limitations:**

See Weakness

**Strengths And Weaknesses:**

Strength:
1. The proposed method is novel, as the standard evaluation protocal is supervised paradigm, an unsupervised paradigm that does not lose performance will be cost-efficient enough.
2. The experiments are extensive ranged, from different datasets, architecture and even diverse SSL objectives.

Weakness:
1. Since the algorithm needs to get the minimum spanning tree, does it mean for every dataset, you need to construct the tree / graph on the representations. Will this be time-consuming? I cannot find any details about how to construct the trees or time complexity of the process.
2. In table 1, for the hyperparameter selection experiments, seems proposed IDEST does not perform well over the baselines in the ResNet representation compared to the outperformance on the ViT-based representations, is this a common situation across other ResNet architectures or solely the VICReg case?

---

> ### Author Rebuttal · Authors · 2026-03-30
>
> We thank you for your feedback and positive comments. Please find below our answers to the weaknesses you raised:
>
> **W1. “Since the algorithm needs to get the minimum spanning tree, does it mean for every dataset, you need to construct the tree / graph on the representations. Will this be time-consuming? I cannot find any details about how to construct the trees or time complexity of the process.”**
>
> For each dataset and model, we compute the dimension estimator $\mathrm{dim}_\mathrm{MST}$. MSTs are computed via Kruskal's algorithm [1] with worst-case complexity $\mathcal{O}(N^2 \cdot (D + \log N))$, where $N$ is the number of samples and $D$ is the embedding dimension. Preprocessing (distance matrix computation and sorting) dominates in practice and is optimized via Ripser [2], which implements an efficient sparse distance pipeline.
>
> IdEst can operate on a subsample of size $N \ll N_D$, where $N_D$ is the full dataset size; sensitivity analyses over 30 models (same as in Fig. 1, see Appendix B.2) show that at $N = 50,000$, Spearman rank correlation reaches $\rho=0.99$, indicating that relative model rankings are highly stable, with only a 3.2% average change in estimated values, justifying this choice.
>
>
> As for the practical runtime, IdEst requires only a single feature-extraction pass followed by the $\mathrm{dim}_\mathrm{MST}$ computation. Table B.1 compares wall-clock time against a 10-epoch linear probe  under a single hyperparameter configuration ($B=1024$, batch size; 2 H100 GPUs) on ImageNet-1k (avg. over 3 runs):
>
>
> #### Table B.1. Computational cost.
>
> | Model | Architecture |  D | Param (M) | Linear Probe (min) | IdEst (min) |
> |:-|:-|:-:|:-:|:-:|:-:|
> | VICReg | ResNet-50 | 2048 | 24 | 64.5 | 2.3 |
> | DINOv2 | ViT-S | 384 | 22 | 65.8 | 1.8 |
> | | ViT-B | 768 | 86 | 64.5 | 2.1 |
> | | ViT-L | 1024 | 303 | 113.3 | 4.9 |
>
> IdEst reduces evaluation time by ${\sim}26{\times}$ on average compared to linear probing (which may require additional hyperparameter tuning, e.g., learning rate selection).
>
>
> **W2 “In table 1, for the hyperparameter selection experiments, seems proposed IDEST does not perform well over the baselines in the ResNet representation compared to the outperformance on the ViT-based representations. Is this a common situation across other ResNet architectures, or solely the VICReg case?”**
>
>
> IdEst's relative underperformance is specific to VICReg, rather than a general limitation on ResNet architectures.
> RankMe and VICReg are inherently aligned: RankMe measures effective rank via the eigenvalue entropy of the embedding matrix, which mirrors VICReg's own variance and covariance regularization terms, making it a particularly favorable setting for RankMe. Moreover, even on VICReg (ResNet-50), IdEst remains close to the ACC-1 upper bound: for instance, on `wd`, IdEst achieves 67.1 against an oracle of 69.1, far from the lower bound of 37.5.
>
> To further validate IdEst’s performance on ResNet, we conducted additional experiments on DINO with a ResNet-50 varying `s-temp` (the student temperature) and the `t-temp` (teacher temperature), and adding the `all` column in which methods must select from the full pool of hyperparameter configurations for a given method. The results are reported in Table B.2.
>
> ##### Table B.2 DINO (RN-50) model selection on ImageNet.
>
> | | **DINO** (ResNet-50) |  |   |
> |:- |:-:|:--:|:-:|
> | **Method**   |`s-temp.`  |  `t-temp.`  |    `all`    |
> | *ACC-1 Bounds* | *[57.9,67.5]* | *[63.0,68.4]* | *[57.9,68.4]* |
> | $\alpha$-ReQ |    61.9     |    63.0     |    63.0     |
> | RankMe |    61.9     |    67.3     |    67.3     |
> |  LiDAR  |   65.5     |    67.3         |    67.3    |
> |  **IdEst** |  **67.5**   |  **67.6**   |  **67.6**   |
>
> IdEst outperforms RankMe and $\alpha$-ReQ across all hyperparameter categories, including the `all` criterion, confirming that IdEst's relative gap on VICReg is not a general limitation on ResNet architectures
>
>
> [1] J.B. Kruskal, On the shortest spanning subtree of a graph and the traveling salesman problem. In Proceedings of the American Mathematical society, 1956
>
> [2] Tralie et al, Ripser. py: A lean persistent homology library for python, JOSS 2018

---

> > ### Author Rebuttal · Reviewer_B9sk · 2026-04-04
> >
> > My questions are fully addressed, thanks for you work.

---

### Official Review · Reviewer_6QvS · 2026-03-15

**Soundness:** 3
**Presentation:** 2
**Significance:** 3
**Originality:** 3
**Overall Recommendation:** 4
**Confidence:** 4

**Summary:**

The authors propose an unsupervised metric to measure performance of self-supervised learning models.  Departing from existing works measuring the effective dimension of the representation space, they propose to instead use the Intrinsic Dimension which can be measured by computing a MST on the representations. The method is shown to be highly effective in practical settings.

**Compliance With Llm Reviewing Policy:**

Affirmed.

**Final Justification:**

The authors adequately answered my concerns during the rebuttal.
My assessment of the work remains positive, and the additional clarifications regarding the algorithmic choices and complexity are appreciated.
Due to the limited improvements over existing work, I will keep my score as Weak Accept.

**Key Questions For Authors:**

No particular question beyond what is mentioned in the weaknesses.

**Limitations:**

Yes

**Strengths And Weaknesses:**

**Strengths:**
- The proposed method is very principled, and motivated by previous theoretical works
- The proposed estimator is compared to previous ones which legitimises its choice over others
- The method performs well in practice as a proxy for classification performance across methods
- Experiments are done at a sufficiently high scale, making the insights applicable in practice

**Weaknesses:**

- Important missing baseline: LiDAR[1]. This method already demonstrated a high correlation to performance especially for I-JEPA and would be important to include for comparison.
- IdEst seems to mainly improve over existing methods for I-JEPA. As it measures something different than previous methods a difference in behaviour is expected, but it does limit the generality of the approach.
- More details on the implementation of the method would benefit the paper. For example, which algorithm was used to compute the length of the MST,  whether IdEst use an exact or approximated algorithm, what is the complexity of the method (both from an algorithmic point of view and practical runtime), or  how are distances calculated to create the MST, since I-JEPA does not give global representations out of the box

**Minor points:**
- Line 027 “withdownstream”, missing space
- The title of the paper should be fixed in the appendix
- SUN397, Aircraft, CUB and CIFAR are mentioned in the paper and appendix but there are no experiments using them in the paper, they should probably be removed.


[1] Thilak, Vimal, et al. "Lidar: Sensing linear probing performance in joint embedding ssl architectures." arXiv preprint arXiv:2312.04000 (2023).

---

> ### Author Rebuttal · Authors · 2026-03-30
>
> We thank you for your feedback and positive comments. Please find below our answers to the weaknesses you raised:
>
>
> **W1 “Important missing baseline: LiDAR[1]. This method already demonstrated a high correlation to performance especially for I-JEPA, and would be important to include for comparison.”**
>
>
> LiDAR leverages SSL pretraining information (positive pairs), while our work targets the harder setting where only frozen representations are accessible, generalizing beyond hyperparameter selection to, e.g., foundation model assessment.
>
> We include LiDAR in Table A.1 for completeness using their `all` column (selection from the full hyperparameter pool), which represents the more challenging configuration. We also add DINO with a ResNet-50 (Table B.2, following R-B9sk). LiDAR is indeed a strong baseline when pretraining details are accessible. Nonetheless, across all four models in the `all` column, IdEst achieves comparable or better model selection than LiDAR (65.5 vs 65.0 on VICReg, 67.6 vs 67.3 on DINO RN-50, 65.5 for both on DINO ViT, 66.4 vs 63.4 on I-JEPA), despite having no access to pretraining information.
>
>
>
> #### **Table A.1 Unsupervised model selection on ImageNet.**
>
> || RN-50 |     |ViT |     |
> |:- |:-:|:-:|:-:|:-:|
> |               | **VICReg**  | **DINO** | **DINO** |  **I-JEPA**  |
> | Method        |     `all`      |    `all`  |    `all`     |       `all`      |
> | *ACC-1 Bounds*|  *[37.5, 69.1]*|  *[57.9,68.4]* | *[48.4, 69.1]* |  *[49.0, 66.5]*  |
> | $\alpha$-ReQ  |      53.5      |  63.0   |     58.6     |        49.0      |
> | RankMe        |    **69.1**    |    67.3  |     63.6     |      61.9      |
> | LiDAR         |      65.0      |    67.3  |   **65.5**   |    63.4 |
> | IdEst         |      65.5      |  **67.6**   |   **65.5**   |   **66.4**    |
>
> Focusing especially on I-JEPA as per your suggestion: IdEst matches or outperforms LiDAR across all three search spaces (Table A.2) and achieves stronger rank correlation with ImageNet Top-1 accuracy ($\rho$=0.87 vs $\rho=0.70$), reflecting better ranking ability.
>
> #### **Table A.2 I-JEPA model selection on ImageNet.**
>
>  |   |  **I-JEPA**  |  |  |  |
> |:-|:-:|:-:|:-:|:-:|
> | Method |  `lr`  |  `target-size`  |  `context-size`  |  `all`  |
> |  *ACC-1 Bounds*   | *[61.9, 66.4]* | *[49.0, 66.4]* | *[61.3, 66.5]* | *[49.0, 66.5]* |
> | $\alpha$-ReQ  |  61.9 | 49.0 | 61.3 | 49.0 |
> | RankMe |  61.9 | 55.9 | 61.3 | 61.9 |
> | LiDAR |  63.4 | **66.4** | **66.0** | 63.4 |
> |  **IdEst**   | **66.4** | **66.4** | **66.0** | **66.4** |
>
>
> **W2 “IdEst seems to mainly improve over existing methods for I-JEPA. As it measures something different than previous methods, a difference in behaviour is expected, but it does limit the generality of the approach.”**
>
>
> We believe IdEst's main benefits stem from its generality: it targets no specific SSL paradigm while providing consistent results.
>
> This consistency is highlighted by the `all` setting (Table A.1 above). IdEst achieves the best or tied-best performance in three out of four configurations, e.g., on I-JEPA (66.4 vs. 63.4 for LiDAR) and DINO ViT (65.5 vs 63.6 for RankMe).
>
> Furthermore, following R-B9sk's remarks (R-B9sk Table B.2), IdEst also recovers most ImageNet oracle performance on DINO (RN-50), outperforming past methods, further showcasing its generalizability. We note that even in configurations where RankMe or $\alpha$-ReQ outperform it, IdEst remains close to the ACC-1 upper bound, unlike competing methods which can exhibit large gaps (e.g., $\alpha$-ReQ's 15.6-point gap on VICReg `wd`, RankMe's 5.5-point gap on DINO ViT `lr`, both exceeding IdEst's largest gap of 4.4 points across all configurations; Table 1). Notably, RankMe's advantage on VICReg can be expected, as its eigenvalue-entropy measure directly mirrors VICReg's own variance-covariance regularization.
>
>
> **W3 “More details on the implementation of the method would benefit the paper. For example, which algorithm was used to compute the length of the MST, whether IdEst use an exact or approximated algorithm, what is the complexity of the method (both from an algorithmic point of view and practical runtime), or how are distances calculated to create the MST, since I-JEPA does not give global representations out of the box.”**
>
> Please refer to our response to R-B9sk (W1 & Table B.1) for details on algorithm, complexity and runtime.
>
> Concerning which representations are used, to satisfy the assumptions of Theorem 3.1, we use the representation passed to the classifier head, following each method's standard evaluation protocol (consistent with how linear probes are trained; Appendix B.1). For models without a class token (e.g., I-JEPA), we average-pool the patch tokens; for models with a `[CLS]` token (e.g., DINO, DINOv2), we use this token directly.
>
> **Minor points** Typos have been corrected. For SUN397, Aircraft, and CUB: these datasets are evaluated in Table 1, under the "fine-grained" rows. We will make this link more explicit in the paper.

---

> > ### Author Rebuttal · Reviewer_6QvS · 2026-04-01
> >
> > Thank you for your answer which addresses my most salient concerns.
> > I had a remaining question, mostly out of curiosity.
> >
> > Regarding the choice of algorithm for the MST computation, I was wondering if you tried different algorithms, and if this would impact the runtime of the method ?
> > - **Kruskal's**: It is best for sparse graphs, but here it seems that the method uses the complete graph
> > - **Prim's**: It can be faster for complete graphs as there is no need to sort all of the edge weights
> > - **Boruvka's**: If the representations are computed across multiple devices, the algorithm can be ran in parallel which may end up more efficient. Implementations are however more complex, and the graphs computed here may be too small to benefit.
> >
> > The method is already fast, so this is more out of curiosity than any practical concers.

---

> > > ### Author Response · Authors · 2026-04-03
> > >
> > > Thank you for raising this point.
> > >
> > > When implementing IdEst, we considered various algorithms for $\mathrm{MST}$ computation. While the asymptotic complexity of Prim's algorithm is better than that of Kruskal’s, our current implementation utilizes Ripser, which is based on Kruskal's algorithm. The reason is that Ripser is highly optimized and effectively bypasses the overhead of a full explicit edge sort, providing an efficient implementation with manageable practical runtime.

---

### Decision · Program_Chairs · 2026-04-30

**Decision:**

Accept (regular)

**Comment:**

The authors introduce an unsupervised metric for assessing the quality of self-supervised learning representations by estimating their intrinsic dimension. Using a Minimum Spanning Tree based estimator, the method provides a geometric perspective that can be more efficient than the supervised metric of linear probing.

The reviewers all found the motivation of the work to be clear with thorough experimental verification, and several appreciated how principled the approach was. Still, there were some issues raised in the initial reviews:
- Lack of detail on time complexity of MST, absence of wall-clock time comparison
- Missing LiDAR baseline
- Performance validated on limited vision architectures
- Novelty of using intrinsic dimension for representation quality
- Generality of vision SSL results to other modalities

The rebuttals and discussion clarified essentially all of these points:
- The MST time complexity is known from prior work, and details were added on how they were computed
- Wall clock time comparisons show MST is favorable over linear probing
- LiDAR and additional CNN architectures were added and do not change conclusions
- Novelty was a point of disagreement among reviewers, but I find no issues with how the paper presents its claims regarding novelty
- The method was not validated on modalities beyond vision, which is the largest outstanding point

Given that the outstanding points are not critical, and sufficient experiments have been done to back up the claims made, I am recommending acceptance. The authors must incorporate the points from the discussion into their final version of the paper.

---

I leave some additional comments for the authors that they can consider at their discretion. For the parametric ID estimator category, MLE and TwoNN are often outperformed by ESS [1] in my experience, so it could make a stronger point of comparison to MST. A convenient implementation of ESS in [2] could be tried. More recently model-based ID estimators have advanced significantly, leveraging today’s powerful generative modelling paradigms, see [3, 4, 5]. The authors can decide whether the model-based literature is relevant enough to discuss.

[1] Johnsson et al. “Low bias local intrinsic dimension estimation from expected simplex skewness” IEEE TPAMI 2014

[2] scikit-dimension, https://github.com/scikit-learn-contrib/scikit-dimension. Bac et al. “Scikit-Dimension: A Python Package for Intrinsic Dimension Estimation” Entropy 2021

[3] Tempczyk et al. “LIDL: Local intrinsic dimension estimation using approximate likelihood” ICML 2022

[4] Stanczuk et al. “Diffusion models encode the intrinsic dimension of data manifolds” ICML 2024

[5] Kamkari et al. “A Geometric View of Data Complexity: Efficient Local Intrinsic Dimension Estimation with Diffusion Models” NeurIPS 2024